# Incomplete Multi-view Clustering via Hierarchical Semantic Alignment and Cooperative Completion

**Xiaojian Ding,**\* **Lin Zhao, Xian Li, Xiaoying Zhu**
School of Computer and Artificial Intelligence,
Nanjing University of Finance and Economics,
Nanjing, China

## Abstract

Incomplete multi-view data, where certain views are entirely missing for some samples, poses significant challenges for traditional multi-view clustering methods. Existing deep incomplete multi-view clustering approaches often rely on static fusion strategies or two-stage pipelines, leading to suboptimal fusion results and error propagation issues. To address these limitations, this paper proposes a novel incomplete multi-view clustering framework based on Hierarchical Semantic Alignment and Cooperative Completion (HSACC). HSACC achieves robust cross-view fusion through a dual-level semantic space design. In the low-level semantic space, consistency alignment is ensured by maximizing mutual information across views. In the high-level semantic space, adaptive view weights are dynamically assigned based on the distributional affinity between individual views and an initial fused representation, followed by weighted fusion to generate a unified global representation. Additionally, HSACC implicitly recovers missing views by projecting aligned latent representations into high-dimensional semantic spaces and jointly optimizes reconstruction and clustering objectives, enabling cooperative learning of completion and clustering. Experimental results demonstrate that HSACC significantly outperforms state-of-the-art methods on five benchmark datasets. Ablation studies validate the effectiveness of the hierarchical alignment and dynamic weighting mechanisms, while parameter analysis confirms the model's robustness to hyperparameter variations. The code is available at
`https://github.com/XiaojianDing/2025-NeurIPS-HSACC`.

## 1 Introduction

Incomplete multi-view data, where certain views are entirely missing for some samples, presents significant challenges for traditional multi-view clustering methods [1]. Due to sensor limitations, occlusions, data acquisition conditions, or storage costs, incomplete multi-view data is widely present in the fields of computer vision, multimedia, and image processing. Since missing views disrupt cross-view correlations, amplify noise interference, and introduce biases, there is a necessity to develop Incomplete Multi-View Clustering (IMVC) methods that jointly address view completion and feature learning to ensure robust clustering performance under incomplete conditions.

Existing IMVC methods can be broadly classified into two categories: traditional approaches and deep learning-based approaches. Traditional IMVC methods usually infer the missing parts based on the available information [2, 3], and subsequently employ specific machine learning techniques for multi-view clustering, such as non-negative matrix factorization methods [4, 5], kernel methods [6, 7], subspace learning methods [8, 9], and graph methods [10, 11]. These shallow IMVC methods commonly suffer from limited linear modeling capacity and sensitivity to complex missing patterns,

---

\*Corresponding author: xjding@nufe.edu.cn

39th Conference on Neural Information Processing Systems (NeurIPS 2025).

hindering their ability to fully exploit latent correlations and complementary information in incomplete multi-view data [7]. Recent advancements in deep learning have led to growing interest in IMVC methods, given their robust generalization ability and high scalability. The commonly used deep IMVC methods include (1) autoencoder-based approaches [12], (2) GAN-based methods [13], (3) GCN-based techniques [14], and (4) contrastive learning-based frameworks [15]. The core of these methods lies in the alignment and fusion of latent representations, which is achieved through various mechanisms. The ultimate goal is to enable effective complementarity and synergy of multi-view information in the latent space. Deep networks can adaptively compensate for missing views by implicitly inferring absent information.

Despite significant advancements in existing deep IMVC methods, these methods still suffer from at least two key limitations. First, traditional methods that rely on static fusion strategies (e.g., uniform weighting) fail to adapt to address the distributional differences between views. This results in suboptimal fusion outcomes due to the inability to dynamically balance the specific contributions of each view. While some approaches employ dynamic fusion strategies (such as calculating view weights based on variance [16] or sharing weights across all views [17]), they lack a hierarchical semantic separation, failing to clearly distinguish between low-level consistency alignment and high-level semantic fusion. Consequently, this often leads to the loss of multi-granularity information, limiting the ability to capture both view-invariant and view-specific features. Second, the conventional two-stage process, where data completion is followed by clustering, suffers from error propagation due to the independent optimization of the completion and clustering objectives. Although some studies have attempted to unify data recovery and clustering into a single framework for joint optimization [18, 19], they overlook the varying significance of different views in the fused representation within specific tasks or distributions. As a result, the completion process fails to fully exploit information from high-quality views, ultimately limiting the overall effectiveness of the method.

To address the aforementioned issues, we propose a novel IMVC framework through Hierarchical Semantic Alignment and Cooperative Completion (HSACC). HSACC first establishes low-level semantic consistency by maximizing mutual information across views, ensuring alignment of shared patterns. In the high-level semantic space, it computes adaptive view weights using distributional affinity between individual views and an initial fused representation, followed by weighted fusion to generate a unified global representation. A discrepancy minimization objective further harmonizes the global representation with view-specific semantics, enhancing cross-view coherence. Then, HSACC implicitly recovers missing views by projecting aligned latent representations into high-dimensional semantic spaces, leveraging learned discriminative features to guide completion. Finally, the completed views and latent representations are jointly optimized through reconstruction and clustering objectives. In summary, our contributions can be summarized as follows:

- We propose a novel hierarchical semantic alignment and dynamic weighted fusion mechanism that, through dual-level semantic space design and adaptive weight allocation, significantly enhances the robustness and discriminability of cross-view fusion.

- We propose a unified framework combining joint optimization and dynamic weighting, which eliminates error propagation in traditional two-stage pipelines by implicitly recovering missing views through discriminative latent representations.

- Experimental results conducted on multiple benchmark datasets demonstrate the superiority of our proposed HSACC over the state of the art IMVC methods.

## 2 Related Work

### 2.1 Deep Incomplete Multi-View Clustering

Leveraging the powerful nonlinear feature extraction capabilities of deep neural networks, deep IMVC approaches have achieved remarkable performance. Existing methods can be systematically categorized into four main groups: (1) **Autoencoder-based methods.** These methods employ shared latent spaces and cross-reconstruction mechanisms during training to capture inter-view dependencies. During inference, they reconstruct the representations of missing views from available ones, thereby effectively mitigating data incompleteness [12, 20, 21]. (2) **GAN-based methods.** By exploiting the strong generative capabilities of Generative Adversarial Networks (GANs), these approaches synthesize features or reconstruct data for missing views from observed view representations. This

strategy preserves the completeness of multi-view representations, with representative works including [13, 22, 23, 24]. (3) **Graph Convolutional Network-based methods.** These methods build graph structures among samples and utilize graph convolution operations to aggregate information from neighboring complete views. Through message propagation, they effectively impute missing view data, as demonstrated in [14, 25, 26]. (4) **Contrastive learning-based methods.** By incorporating contrastive loss to maximize agreement between positive pairs across different views, these approaches leverage complementary inter-view information to infer representations for missing views [15, 18, 27].

## 2.2 Contrastive Learning

The core principle of contrastive learning [28, 29, 30] is to learn robust feature representations by contrasting similarities between positive and negative samples. This paradigm is particularly effective for incomplete multi-view data, as it enhances representation robustness through cross-view alignment. For instance, PVC-SSN [15] addresses incomplete multi-view clustering via self-supervised contrastive learning, where a self-supervised module enhances the discriminative capacity of the learned representations. Similarly, COMP [18] achieves both cross-view consistency and missing view recovery by maximizing mutual information across views while minimizing conditional entropy. Prolmp [27] introduces a prototype-based imputation framework that learns view-specific prototypes and sample relationships to recover missing view data, while contrastive learning is employed to strengthen both sample and prototype representations. The ADCL method [31] proposes a direct contrastive learning scheme by performing contrastive alignment on sub-vectors of latent features, effectively preventing dimensional collapse. Finally, MICA [32] explores the internal dependencies of multi-view data through a multi-level imputation strategy combined with instance- and cluster-level contrastive learning, further improving representation integrity under incomplete settings.

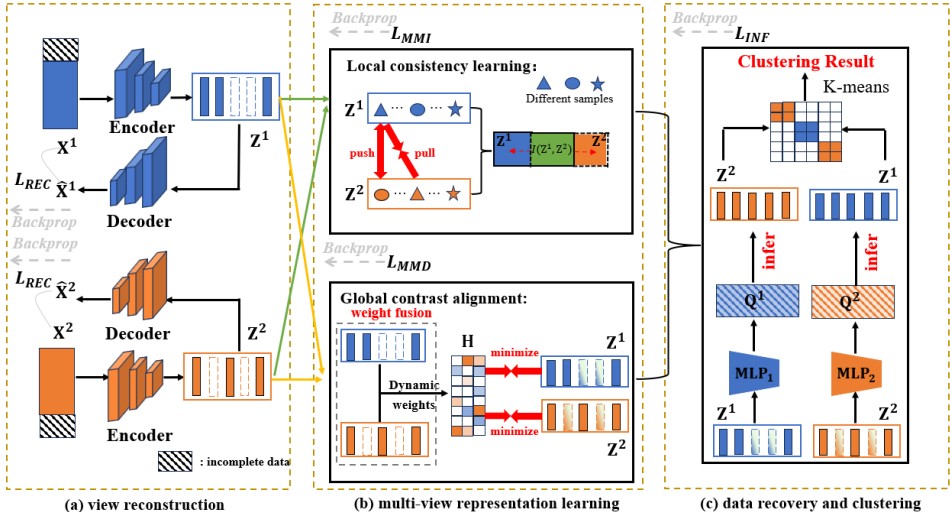

Figure 1: The framework consists of three modules: (a) view reconstruction, where autoencoders extract latent representations and reconstruct the inputs; (b) multi-view representation learning, which aligns views by maximizing mutual information, learns adaptive weighting to fuse complementary information, and minimizes distribution discrepancies; (c) data recovery and clustering, where missing views are completed and clustering is performed based on the completed representations.

## 3 Method

As shown in Figure 1, given two views $\mathbf{X}^1$ and $\mathbf{X}^2$, we first map them to $\mathbf{Z}^1$ and $\mathbf{Z}^2$ via autoencoders, and then extract semantic information at two levels. In the low-level semantic space, the model maximizes the mutual information between views to ensure consistency alignment. In the high-level semantic space, $\mathbf{Z}^1$ and $\mathbf{Z}^2$ are concatenated to form the initial fused representation $\mathbf{R}$, and cross-view distribution alignment is introduced to measure the matching degree between each view representation

$\mathbf{Z}^v$ and the fused representation $\mathbf{R}$, to adaptively estimate view-specific weights. Then, using the computed weights, a weighted fusion of all view representations is performed to obtain a unified high-level shared representation $\mathbf{H}$, and an alignment objective is further designed to minimize the distribution discrepancy between $\mathbf{H}$ and each view. Furthermore, based on the current learned latent representations $\mathbf{Z}^1$ and $\mathbf{Z}^2$, the model maps them to a high-dimensional semantic space through an MLP to obtain $\mathbf{Q}^1$ and $\mathbf{Q}^2$, and uses these results to complete the missing views, obtaining the complete view representations for subsequent clustering. The joint optimization of representation learning and data completion mutually reinforces both processes, thereby improving the overall clustering performance.

## 3.1 Notation

In this paper, $\left\{\mathbf{X}^v \in \mathbb{R}^{N \times d_v}\right\}_{v=1}^V$ represents a multi-view data set with $V$ views, where $\mathbf{X}^v = \{\mathbf{x}_1^v, ..., \mathbf{x}_l^v, ..., \mathbf{x}_N^v\}$, $d_v$ represents the feature dimension of the $v$-th view. Without loss of generality, we consider a two-view dataset as an example, where $V = 2$. Here, $N$ denotes the total number of samples, and $\mathbf{x}_i^v$ indicates the $i$-th sample in the $v$-th view. Additionally, we define an indicator matrix $\mathbf{A} \in \{0,1\}^{N \times V}$, whose elements are defined as follows:

$$\mathbf{A}_{iv} = \begin{cases} 1, & \text{if the } i\text{-th sample is available in the } v\text{-th view} \\ 0, & \text{otherwise.} \end{cases} \tag{1}$$

In this matrix, each column represents the availability status of all instances in a specific view.

## 3.2 The Loss Function

**View Reconstruction Loss**   We construct view-specific autoencoders, where $E^v$ and $D^v$ denote the encoder and decoder for the $v$-th view, respectively. Given the input $\mathbf{X}^v$, the encoder extracts the latent representation $\mathbf{Z}^v = E^v(\mathbf{X}^v) \in \mathbb{R}^{N \times D}$, which is then reconstructed through the decoder as $\hat{\mathbf{X}}^v = D^v(\mathbf{Z}^v)$. The reconstruction loss is computed using Mean Squared Error (MSE), which enforces the preservation of essential structural information in the latent space. The overall reconstruction loss is defined as follows:

$$L_{REC} = \sum_v \|\mathbf{X}^v - D^v(E^v(\mathbf{X}^v))\|^2. \tag{2}$$

**Cross-view Consistency Loss**   In deep multi-view learning, maximizing the mutual information between features of different views is crucial for improve representation consistency [18]. To this end, we estimate mutual information via feature-level similarity. This approach explicitly models the joint and marginal distributions of the features from different views. Let the latent representations of the $i$-th sample in view 1 and view 2 be $\mathbf{z}_i^1$ and $\mathbf{z}_i^2$, respectively, where $z_{i,m}^1$ and $z_{i,n}^2$ represent the $m$-th and $n$-th scalar features. The similarity between the features of this sample across the two views is defined as $p_{m,n}^{(i)} = z_{i,m}^1 \cdot z_{i,n}^2$. By aggregating the similarities across all samples, we obtain the joint probability distribution $\mathbf{P} \in \mathbb{R}^{D \times D}$ between view 1 and view 2. The $(m,n)$-th element of the matrix is defined as:

$$P_{(m,n)} = \frac{1}{N} \sum_{i=1}^N p_{m,n}^{(i)} = \frac{1}{N} \sum_{i=1}^N z_{i,m}^1 \cdot z_{i,n}^2. \tag{3}$$

By summing over the rows and columns of the joint probability distribution $\mathbf{P} \in \mathbb{R}^{D \times D}$, we obtain the marginal probability distributions $\mathbf{P}^{(1)}, \mathbf{P}^{(2)} \in \mathbb{R}^D$ for each view, with elements defined as $P_{(m)} = \sum_{n=1}^D P_{(m,n)}$ and $P_{(n)} = \sum_{m=1}^D P_{(m,n)}$. Based on the above distributions, we define the mutual information loss between views as:

$$L_{MMI} = -I(\mathbf{Z}^1; \mathbf{Z}^2) = -\sum_{m=1}^D \sum_{n=1}^D P_{(m,n)} \ln \frac{P_{(m,n)}}{P_{(m)} \cdot P_{(n)}}, \tag{4}$$

where $I(\mathbf{Z}^1; \mathbf{Z}^2)$ measures the information correlation between $\mathbf{Z}^1$ and $\mathbf{Z}^2$. Minimizing this loss is equivalent to maximizing their mutual information.

**Distribution Alignment Loss**    In this section, we use the Maximum Mean Discrepancy (MMD) to simultaneously calculate view weights and optimize feature representations. Specifically, We first measure the distribution difference between each view and the initial fusion representation to assign weights, then perform weighted fusion to obtain a higher-level common representation and minimize its discrepancy with each view, thereby enhancing the consistency and complementarity of cross-view representations.

*Step 1*: **Estimating View Weights**

For the given two view representations $\mathbf{Z}^1$ and $\mathbf{Z}^2$, we first calculate their initial fusion representation $\mathbf{R}$ as $\mathbf{R} = \frac{\mathbf{Z}^1 + \mathbf{Z}^2}{2}$. Subsequently, to evaluate the contribution of each view, we compute the distribution discrepancy between each view and the initial fused representation $\mathbf{R}$ based on similarity measures. Under the assumption of a linear kernel (i.e., $k(x, y) = x^T y$), we calculate the internal similarity of each view, the internal similarity of the fused representation, and their mutual similarity via dot products. This allows us to quantify the discrepancy between the view-specific representation and the initial fused representation, as follows:

$$D(\mathbf{Z}^v, \mathbf{R}) = \frac{1}{N^2} \left( \sum_{i=1}^{N} \sum_{j=1}^{N} \mathbf{z}_i^v \cdot \mathbf{z}_j^v + \sum_{i=1}^{N} \sum_{j=1}^{N} \mathbf{r}_i \cdot \mathbf{r}_j - 2 \sum_{i=1}^{N} \sum_{j=1}^{N} \mathbf{z}_i^v \cdot \mathbf{r}_j \right), \quad (5)$$

where $N$ is the number of samples, $\mathbf{z}_i^v$ and $\mathbf{r}_i$ represent the $i$-th sample of the $v$-th view and the initial fusion representation $\mathbf{R}$, respectively.

Considering that a view whose latent representation is highly consistent with the initial fused representation should contribute more to the final decision, we assign it a higher weight. Conversely, a view exhibiting a greater discrepancy is assigned a lower weight. This design enables the dynamic adjustment of each view's contribution. Accordingly, we define a weight function $W^v = f(D(\mathbf{Z}^v, \mathbf{R}))$ to calculate the weights, where $f(D)$ is defined as:

$$f(D) = \frac{\exp\left(-D(\mathbf{Z}^v, \mathbf{R})\right)}{\sum_{v=1}^{V} \exp\left(-D(\mathbf{Z}^v, \mathbf{R})\right)}. \quad (6)$$

In multi-view learning tasks, since each view $\mathbf{Z}^v$ may carry complementary information, we perform weighted fusion of all view-specific representations using the learned weights $W^v$, in order to integrate diverse perspectives into the high-level common representation $\mathbf{H}$, where $\mathbf{H} = \sum_{v=1}^{V} W^v \cdot \mathbf{Z}^v$.

*Step 2*: **Distribution alignment between $\mathbf{H}$ and $\mathbf{Z}^v$**

Motivated by domain adaptation [33], to further enhance the information interaction between views and promote the completion of missing views, we aim to align the distribution of single-view representations $\mathbf{Z}^v$ with the high-level common representation $\mathbf{H}$. Specifically, let $\mathcal{P}$ denote the distribution of the view-specific representations $\mathbf{Z}^v = \{\mathbf{z}_1^v, \mathbf{z}_2^v, \ldots, \mathbf{z}_i^v\}_{i=1}^N$, and let $\mathcal{Q}$ denote the distribution of the high-level common representations $\mathbf{H} = \{\mathbf{h}_1, \mathbf{h}_2, \ldots, \mathbf{h}_j\}_{j=1}^N$. We measure their discrepancy in the high-level semantic space by computing the distance between $\mathcal{P}$ and $\mathcal{Q}$:

$$L_{MMD} = \sum_{v=1}^{V} \text{MMD}^2(\mathcal{P}_{\mathbf{Z}^v}, \mathcal{Q}_{\mathbf{H}}) = \sum_{v=1}^{V} \|\mathbb{E}_{x \sim \mathcal{P}_{\mathbf{Z}^v}}[\phi(x)] - \mathbb{E}_{y \sim \mathcal{Q}_{\mathbf{H}}}[\phi(y)]\|_{\mathcal{H}}^2. \quad (7)$$

Here, $L_{MMD}$ denotes the overall discrepancy loss between all views and the high-level common representation $\mathbf{H}$, where $\mathcal{H}$ is the Reproducing Kernel Hilbert Space (RKHS). The terms $\mathbb{E}_{x \sim \mathcal{P}_{\mathbf{Z}^v}}[\phi(x)]$ and $\mathbb{E}_{y \sim \mathcal{Q}_{\mathbf{H}}}[\phi(y)]$ represent the mean embeddings of the $v$-th view and the common representation $\mathbf{H}$ in the RKHS, respectively. Here, $\phi(\cdot)$ is a feature mapping function that projects samples into the RKHS. Since it is difficult to compute the expectations directly, we approximate them using sample means. Taking $\mathbb{E}_{x \sim \mathcal{P}_{\mathbf{Z}^v}}[\phi(x)]$ as an example:

$$\mathbb{E}_{x \sim \mathcal{P}_{\mathbf{Z}^v}}[\phi(x)] \approx \frac{1}{N} \sum_{i=1}^{N} \phi(\mathbf{z}_i^v) = \frac{1}{N} \mathbf{1}_N^T \Phi(\mathbf{Z}^v), \quad (8)$$

where $\Phi(\mathbf{Z}^v) = [\phi(\mathbf{z}_1^v)^{\mathsf{T}}, \phi(\mathbf{z}_2^v)^{\mathsf{T}}, \ldots, \phi(\mathbf{z}_N^v)^{\mathsf{T}}]^{\mathsf{T}}$ denote the feature mapping of all samples in view $\mathbf{Z}^v$, where $\mathbf{1}_N$ is a vector of length $N$ with all elements equal to 1. Similarly, the expectation $\mathbb{E}_{y \sim \mathcal{Q}_{\mathbf{H}}}[\phi(y)]$ can be approximated. In RKHS, the inner product of feature mappings can be computed using the kernel function $k(x, y) = \langle \phi(x), \phi(y) \rangle_{\mathcal{H}}$. To simplify the computation, we construct kernel matrices $\mathbf{K}_{\mathbf{Z}^v \mathbf{Z}^v}$, $\mathbf{K}_{\mathbf{HH}}$, and $\mathbf{K}_{\mathbf{Z}^v \mathbf{H}}$, each of size $N \times N$, representing the internal similarities of

the view-specific representations, the internal similarities of the high-level common representation, and the similarities between them, respectively. By substituting Eq. (8) into Eq. (7) and expanding it using kernel matrices, we obtain the final distribution alignment loss:

$$L_{MMD} = \sum_{v=1}^{V} \frac{1}{N^2} \mathbf{1}_N^T \mathbf{K}_{\mathbf{Z}^v \mathbf{Z}^v} \mathbf{1}_N + \frac{1}{N^2} \mathbf{1}_N^T \mathbf{K}_{\mathbf{HH}} \mathbf{1}_N - \frac{2}{N^2} \mathbf{1}_N^T \mathbf{K}_{\mathbf{Z}^v \mathbf{H}} \mathbf{1}_N. \tag{9}$$

**Inference Consistency Loss** In incomplete multi-view clustering, we design cross-view inference modules $f_{\text{MLP1}}$ and $f_{\text{MLP2}}$ to learn nonlinear mappings in the latent space for implicitly completing missing views. Specifically, the latent representations $\mathbf{Z}^1$ and $\mathbf{Z}^2$ are projected into a high-dimensional semantic space, yielding $\mathbf{Q}^1 = f_{\text{MLP1}}(\mathbf{Z}^1) = f_{\text{MLP1}}(E^1(\mathbf{X}^1))$ and $\mathbf{Q}^2 = f_{\text{MLP2}}(\mathbf{Z}^2) = f_{\text{MLP2}}(E^2(\mathbf{X}^2))$, which leverage information from the complete views to infer the semantics of missing views without relying on explicit interpolation or generation. To measure the semantic consistency between the inferred results and the original latent representations, we introduce the inference loss $L_{INF}$ to supervise the accuracy of the cross-view mapping:

$$L_{INF} = \frac{1}{N} \sum_{i=1}^{N} \left\| \mathbf{z}_i^2 - \mathbf{q}_i^1 \right\|_2^2 + \frac{1}{N} \sum_{i=1}^{N} \left\| \mathbf{z}_i^1 - \mathbf{q}_i^2 \right\|_2^2. \tag{10}$$

Here, $\mathbf{z}_i^v$ is the true latent representation of the $i$-th sample in the $v$-th view, and $\mathbf{q}_i^v$ is the inference result obtained by the inference module for the corresponding view. By minimizing this loss, effective inference between views can be achieved, completing better completion.

**Extending to Multi-View** For $V$ views, we first construct independent autoencoders for each view to extract latent representations, and compute the cross-view consistency loss over all view pairs $(v_1, v_2)$ as $L_{MMI} = \frac{2}{V(V-1)} \sum_{v_1 < v_2} L_{MMI}(\mathbf{Z}^{v_1}, \mathbf{Z}^{v_2})$, after which the latent representations are fused to obtain the global representation $\mathbf{H}$, followed by distribution alignment with each view through the loss $L_{MMD}$. Finally, in cross-view inference, the latent representation of view $v_1$ is projected by an MLP to infer the semantics of view $v_2$, and the inference consistency loss is defined as $L_{INF} = \frac{1}{N} \sum_{i=1}^{N} \frac{1}{V(V-1)} \sum_{v_1 \neq v_2} \left\| \mathbf{z}_i^{v_2} - \mathbf{q}_i^{v_1 \to v_2} \right\|_2^2$, to ensure consistency between the inferred results and the true latent representations.

### 3.3 Objective Function and Optimization Algorithm

Thus, our overall objective function can be expressed as:

$$L = \lambda_1 L_{REC} + \lambda_2 L_{INF} + \lambda_3 L_{MMI} + \lambda_4 L_{MMD}. \tag{11}$$

The $\lambda_1$, $\lambda_2$, $\lambda_3$, and $\lambda_4$ are trade-off parameters in the loss function. Ultimately, we optimize Eq. (11) to recover the missing views, thereby obtaining complete view representations. These representations are then concatenated and subjected to $k$-means clustering to obtain the final clustering results. The overall training procedure of HSACC is summarized in Algorithm 1.

---

**Algorithm 1:** Incomplete Multi-View Clustering via Hierarchical Semantic Alignment and Cooperative Completion

---

**Input:** Incomplete multi-view dataset $\{\mathbf{X}^v\}_{v=1}^{V}$ with indicator matrix $\mathbf{A}$, epoch $E$, $E_1$ ;
Parameters: Trade-off coefficients $\lambda_1, \lambda_2, \lambda_3, \lambda_4$
**Output:** Clustering results

**for** $epoch = 1$ **to** $E$ **do**
    Learn view-specific representations via Eqs. (2) and (4);
    Concatenate the learned $\mathbf{Z}^v$ to obtain $\mathbf{R}$;
    Calculate the view weight $W^v$ using Eq. (6);
    Fuse based on the weight $W^v$ to obtain $\mathbf{H}$;
    Calculate the distribution discrepancy between $\mathbf{H}$ and $\mathbf{Z}^v$ via Eq. (9);
    **if** $epoch \geq E_1$ **then**
        Obtain all $\mathbf{Q}^{v_1 \to v_2}$ via cross-view inference MLP to complete the views;
        Update representations via joint optimization using Eq. (11);

Perform $k$-means clustering algorithm on the concatenated views;

---

# 4 Experiments

## 4.1 Experimental Setting

**Datasets**  To evaluate the effectiveness of the proposed method, we selected five representative datasets. LandUse_21 [34] contains 2,100 remote sensing images from 21 categories. Noisy MNIST [35] consists of noisy handwritten digit images, with the original images as View 1 and Gaussian-noised images as View 2. Caltech101-20 [36] contains 2,386 images from 20 categories using HOG and GIST features. Hdigit [37] contains 10,000 handwritten digit images from 10 categories. 100leaves [38] contains 1,600 samples from 100 categories.

**Comparing Methods**  We conducted comparative experiments with nine state-of-the-art incomplete multi-view clustering methods. **RPCIC** [39] addresses the IMVC problem through cross-view contrastive learning and robust prototype discriminative learning. **MRL_CAL** [19] tackles IMVC by combining contrastive learning with adversarial learning, aiming to learn multi-level features to enhance clustering performance. **ICMVC** [14] leverages contrastive learning with high-confidence guidance to handle incomplete multi-view data. **MCAC** [40] employs attention-based contrastive learning to enhance consistent representation across views and effectively handle missing data. **PROLMP** [27] restores missing views by learning view-specific prototypes and instance-prototype relationships, and further enhances feature representation via contrastive learning. **DCP** [41] adopts dual contrastive prediction and an information-theoretic framework to achieve both data recovery and view-consistency learning. **DSIMVC** [42] dynamically completes missing views through a bi-level optimization framework and automatically selects high-quality imputed samples for training. **SURE** [43] handles partially missing samples in IMVC via noise-robust contrastive learning and a class-level identification framework. **COMP** [18] simultaneously achieves data recovery and view-consistency learning by integrating contrastive learning and a dual-prediction mechanism.

**Experimental Configuration**  All experiments were conducted on an NVIDIA RTX 4070 GPU using PyTorch 2.3.1. Our method employs a unified fully connected autoencoder, with the encoder structured as Input–1024–1024–1024–Output and the decoder mirrored accordingly. During the data recovery phase, a multilayer perceptron (MLP) with hidden layer dimensions of 256–128–256 is used for cross-view inference and feature reconstruction. In our experiments, each dataset underwent $E$ training epochs (e.g., $E = 500$), and the computation of the inference loss was introduced starting from the $E_1$-th epoch (e.g., $E_1 = 100$). The learning rate was set to 0.0001, and the batch size was 256. We conducted experiments under different missing rates to evaluate the performance of various comparative methods.

## 4.2 Experimental Results and Analysis

**Performance Comparison**  Table 1 presents the clustering results of various IMVC methods under different missing rates, where the best and second-best results are highlighted in bold and underline, respectively. It can be observed that our method consistently outperforms the other nine approaches across all datasets. Among them, DSIMVC and DCP are two state-of-the-art imputation-based IMVC methods. However, the experimental results demonstrate that our method still achieves superior performance compared to both. In particular, on the Caltech101-20 dataset with a missing rate of 0.5, HSACC improves the ACC and ARI metrics by 5.3% and 8.57%, respectively, compared to the second-best method. Moreover, on the Noisy MNIST dataset, when the missing rate increases from 0.3 to 0.7, the accuracy of HSACC decreases by only 6.92%, whereas ICMVC suffers a drop of 35.19%. These results indicate that HSACC is capable of accurately capturing the intrinsic structural features within each view from incomplete multi-view data and effectively reconstructing the missing parts using the learned representations, thereby exhibiting excellent robustness and generalization ability.

**Ablation Study**  To evaluate the effectiveness of each loss component, we conducted ablation experiments on the Caltech101-20 dataset, focusing on four modules: $L_{REC}$, $L_{MMI}$, $L_{MMD}$, and $L_{INF}$. As shown in Table 2, the results indicate that removing any of these components leads to a significant drop in model performance. Specifically, the baseline model M-1, which only includes $L_{REC}$, achieves an ACC, NMI, and ARI of 42.07%, 41.06%, and 28.38%, respectively. By introducing the cross-view consistency loss $L_{MMI}$, model M-9 shows substantial improvements

Table 1: Performance comparison under different missing rates on five datasets.

| Missing Rate | Method | LandUse_21 | | | Noisy MNIST | | | Caltech101-20 | | | Hdigit | | | 100leaves | | |
|---|---|---|---|---|---|---|---|---|---|---|---|---|---|---|---|---|
| | | ACC | NMI | ARI | ACC | NMI | ARI | ACC | NMI | ARI | ACC | NMI | ARI | ACC | NMI | ARI |
| 0.3 | RPCIC | 21.29 | 26.52 | 8.21 | 53.19 | 52.18 | 39.21 | 41.49 | 58.30 | 34.97 | 91.06 | 82.49 | 81.23 | 53.62 | 76.32 | 38.20 |
| | MRL_CAL | 19.76 | 21.21 | 6.85 | 15.73 | 2.80 | 1.18 | 20.03 | 38.25 | 12.54 | 12.30 | 0.69 | 0.23 | 12.88 | 50.63 | 7.31 |
| | ICMVC | 26.14 | 30.01 | 13.13 | 95.79 | 91.90 | 93.07 | 35.88 | 61.16 | 26.65 | 16.34 | 8.64 | 3.56 | 20.49 | 76.62 | 30.88 |
| | MCAC | 20.21 | 21.50 | 7.08 | 95.17 | 91.36 | 91.99 | 32.62 | 46.97 | 22.64 | 29.49 | 20.02 | 11.40 | 35.26 | 64.88 | 20.58 |
| | PROLMP | 20.33 | 21.81 | 8.37 | 88.80 | 78.47 | 76.88 | 32.47 | 53.08 | 25.33 | 93.17 | 87.46 | 84.81 | 51.54 | 77.34 | 38.29 |
| | DCP | 26.84 | 30.40 | 13.84 | 72.98 | 77.22 | 58.93 | 71.51 | 70.53 | 79.06 | 94.31 | 93.33 | 89.43 | 32.21 | 75.34 | 29.62 |
| | DSIMVC | 21.43 | 25.25 | 8.04 | 63.70 | 60.70 | 51.40 | 27.57 | 49.72 | 21.03 | 94.30 | 91.38 | 91.30 | 29.78 | 61.91 | 17.53 |
| | SURE | 24.80 | 29.04 | 11.10 | 95.11 | 91.05 | 91.88 | 49.70 | 65.26 | 40.96 | 49.74 | 38.18 | 23.94 | 46.51 | 71.21 | 30.04 |
| | COMP | 26.85 | 31.50 | 12.87 | 84.76 | 81.89 | 77.00 | 71.55 | 69.61 | 79.00 | 93.84 | 85.66 | 86.86 | 38.39 | 72.15 | 28.00 |
| | **Ours** | **27.39** | **31.80** | **14.27** | **95.81** | **92.33** | **93.80** | **72.34** | **71.48** | **79.36** | **94.89** | **93.78** | **92.06** | **54.28** | **77.57** | **39.21** |
| 0.5 | RPCIC | 19.24 | 22.49 | 5.83 | 51.63 | 47.77 | 35.75 | 46.19 | 62.70 | 38.53 | 83.58 | 73.18 | 66.95 | 40.13 | 71.69 | 27.63 |
| | MRL_CAL | 18.52 | 21.16 | 6.29 | 14.98 | 3.20 | 1.22 | 31.52 | 38.24 | 24.25 | 12.62 | 0.82 | 0.28 | 12.12 | 49.35 | 7.03 |
| | ICMVC | 24.76 | 27.20 | 11.29 | 86.84 | 82.33 | 80.00 | 33.70 | 58.22 | 25.06 | 16.44 | 8.67 | 3.64 | 51.39 | 74.71 | 33.83 |
| | MCAC | 17.91 | 18.52 | 6.14 | 91.03 | 84.21 | 84.98 | 32.56 | 33.17 | 16.78 | 24.08 | 14.75 | 6.93 | 27.59 | 58.43 | 12.14 |
| | PROLMP | 18.26 | 19.75 | 6.41 | 79.01 | 85.11 | 83.12 | 33.16 | 52.87 | 24.60 | 86.05 | 78.45 | 68.16 | 39.86 | 69.78 | 26.56 |
| | DCP | 27.34 | 31.56 | 14.34 | 86.50 | 81.21 | 78.26 | 71.51 | 71.60 | 78.56 | 94.10 | 86.28 | 87.40 | 35.50 | 74.68 | 30.33 |
| | DSIMVC | 21.53 | 25.29 | 8.34 | 83.93 | 76.72 | 73.95 | 28.18 | 48.94 | 21.00 | 94.40 | 89.37 | 90.40 | 27.29 | 59.14 | 15.10 |
| | SURE | 25.68 | 30.45 | 11.99 | 92.34 | 84.99 | 84.31 | 48.78 | 58.52 | 42.04 | 66.59 | 17.66 | 15.47 | 30.52 | 58.81 | 10.46 |
| | COMP | 25.24 | 31.69 | 13.53 | 78.68 | 74.28 | 67.37 | 70.13 | 69.04 | 76.10 | 91.36 | 80.73 | 81.85 | 28.40 | 62.32 | 15.68 |
| | **Ours** | **28.11** | **31.89** | **14.60** | **92.55** | **85.92** | **85.50** | **76.81** | **73.15** | **87.13** | **95.57** | **90.68** | **90.61** | **52.17** | **74.78** | **35.30** |
| 0.7 | RPCIC | 17.71 | 18.73 | 4.45 | 47.24 | 43.19 | 30.47 | 41.53 | 57.32 | 38.94 | 49.63 | 42.52 | 31.86 | 35.19 | 67.95 | 21.22 |
| | MRL_CAL | 16.14 | 16.83 | 4.92 | 15.37 | 2.42 | 1.01 | 24.43 | 30.13 | 13.61 | 12.59 | 0.92 | 0.38 | 9.94 | 44.78 | 4.66 |
| | ICMVC | 20.71 | 23.79 | 8.33 | 60.60 | 62.62 | 49.92 | 32.51 | 58.04 | 25.32 | 16.53 | 8.94 | 3.67 | 43.43 | 67.84 | 25.21 |
| | MCAC | 17.20 | 18.52 | 5.03 | 82.91 | 73.41 | 70.02 | 41.08 | 42.96 | 36.68 | 24.78 | 16.48 | 7.55 | 27.96 | 58.19 | 11.35 |
| | PROLMP | 14.74 | 14.68 | 3.68 | 59.23 | 46.45 | 34.17 | 31.77 | 50.79 | 23.29 | 82.97 | 73.36 | 63.36 | 34.27 | 65.00 | 19.27 |
| | DCP | 24.89 | 27.04 | 11.27 | 85.03 | 78.10 | 77.12 | 70.76 | 69.80 | 75.23 | 90.98 | 84.02 | 85.12 | 34.98 | 67.86 | 24.96 |
| | DSIMVC | 20.87 | 23.69 | 7.76 | 58.10 | 55.50 | 44.30 | 27.33 | 46.77 | 19.98 | 91.05 | 85.67 | 84.67 | 25.12 | 56.97 | 12.84 |
| | SURE | 25.36 | 28.74 | 11.12 | 83.12 | 77.63 | 74.08 | 38.86 | 53.08 | 25.77 | 44.50 | 31.52 | 22.56 | 21.90 | 52.76 | 6.93 |
| | COMP | 24.25 | 29.11 | 10.43 | 68.72 | 65.59 | 55.88 | 69.28 | 67.58 | 75.04 | 87.45 | 73.86 | 74.29 | 34.53 | 66.85 | 24.23 |
| | **Ours** | **27.56** | **30.41** | **13.85** | **88.89** | **79.05** | **77.73** | **72.02** | **70.39** | **76.30** | **91.50** | **85.86** | **85.91** | **43.47** | **68.49** | **25.99** |

across all three metrics, with an ACC of 68.69%, NMI of 67.63%, and ARI of 73.68%. By further incorporating the distribution alignment loss $L_{MMD}$ in model M-12, the ACC increases to 71.29%, and both NMI and ARI also improve correspondingly. When all four loss modules are utilized (M-15), the model achieves the highest performance, with all metrics surpassing those of models using only partial components. This clearly demonstrates that each loss module plays a crucial role in improving the model's performance.

### 4.3 Model Analysis

**Parameter Analysis**  In this section, we analyze the sensitivity of the four hyperparameters $\lambda_1$, $\lambda_2$, $\lambda_3$, and $\lambda_4$ in the total loss function of our proposed method. Experiments are conducted on the Caltech101-20 dataset with a missing rate of 0.5, and each hyperparameter is evaluated over the range $\{0.01, 0.1, 1, 10, 100\}$. As shown in Figure 2, smaller values of $\lambda_1$ and $\lambda_2$ lead to better performance, with the optimal value being 0.1, while the best values for $\lambda_3$ and $\lambda_4$ are 10 and 1, respectively. Although varying these hyperparameters has some effect on model performance, the fluctuations are minor, indicating that the model is relatively robust to hyperparameter changes.

**Visualization**  As shown in Figure 3, we perform t-SNE visualizations of the learned common representations on the Caltech101-20 and Noisy MNIST datasets. Figures (a)–(d) and (e)–(h) show the clustering results at different training epochs for the two datasets, respectively. As training progresses, the clustering structures in the feature space become increasingly clear, with more distinct inter-class boundaries and more compact intra-class distributions, demonstrating stronger discriminative ability and improved clustering performance.

**Convergence Analysis**  To verify the convergence of the proposed method, we conduct experiments on multiple datasets to observe the changes in loss values and clustering metrics over training epochs. As shown in Figure 4, the corresponding curves on the Caltech101-20 and Noisy MNIST and LandUse_21 datasets indicate that as the number of training epochs increases, the loss values decrease

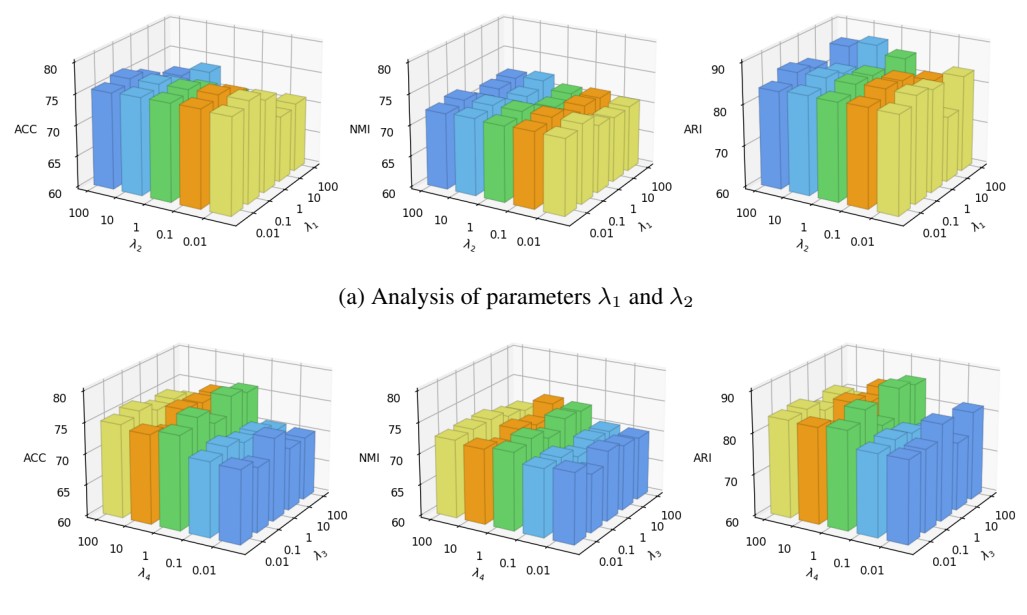

(a) Analysis of parameters $\lambda_1$ and $\lambda_2$

(b) Analysis of parameters $\lambda_3$ and $\lambda_4$

Figure 2: Parameter sensitivity analysis on Caltech101-20 dataset with missing rate 0.5.

Table 2: Performance comparison across different loss combinations on Caltech101-20 dataset.

| Model | M-1 | M-2 | M-3 | M-4 | M-5 | M-6 | M-7 | M-8 | M-9 | M-10 | M-11 | M-12 | M-13 | M-14 | M-15 |
|---|---|---|---|---|---|---|---|---|---|---|---|---|---|---|---|
| $L_{REC}$ | ✓ | | | | ✓ | ✓ | | ✓ | | | ✓ | ✓ | | ✓ | ✓ |
| $L_{MMI}$ | | ✓ | | | | | | | ✓ | ✓ | ✓ | | ✓ | ✓ | ✓ |
| $L_{MMD}$ | | | ✓ | | ✓ | ✓ | | ✓ | | | ✓ | ✓ | ✓ | | ✓ |
| $L_{INF}$ | | | | ✓ | ✓ | | ✓ | | ✓ | ✓ | | ✓ | ✓ | ✓ | ✓ |
| ACC | 42.07 | 54.92 | 36.90 | 43.96 | 42.45 | 51.52 | 55.97 | 67.40 | 68.69 | 71.60 | 50.80 | 71.29 | 72.31 | 73.15 | 76.81 |
| NMI | 41.06 | 54.70 | 44.70 | 31.71 | 28.38 | 53.68 | 57.85 | 67.97 | 67.63 | 71.12 | 53.12 | 68.48 | 71.65 | 71.25 | 73.15 |
| ARI | 28.38 | 51.66 | 23.50 | 29.61 | 31.05 | 38.05 | 51.22 | 71.08 | 73.68 | 80.30 | 34.68 | 78.27 | 78.72 | 79.88 | 87.13 |

significantly, while the clustering metrics steadily improve. These results further demonstrate the convergence and stability of the proposed method across different datasets.

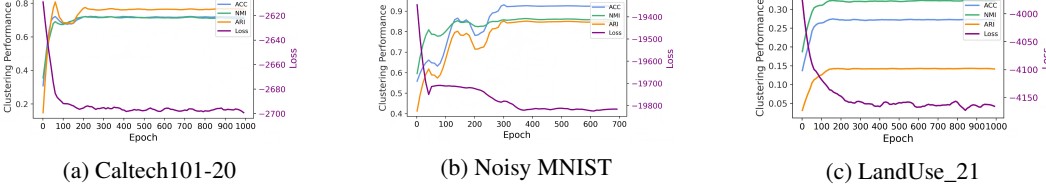

(a) Caltech101-20        (b) Noisy MNIST        (c) LandUse_21

Figure 4: Clustering metrics (ACC, NMI, ARI) and loss curves over training epochs on the Caltech101-20, Noisy MNIST, and LandUse_21 datasets.

## 4.4 Experiments With More Than Two Views

To validate scalability, we will add experiments on 5-view Mfeat dataset. Table 3 compares the training time and clustering metrics (ACC, NMI, ARI) of multi-view clustering with different numbers of views (from 2 to 5). As the number of views increases, the training time rises from 52.80 seconds for 2 views to 186.99 seconds for 5 views, while the clustering performance steadily improves, achieving the highest ACC (88.64), NMI (88.14), and ARI (81.09) with 5 views.

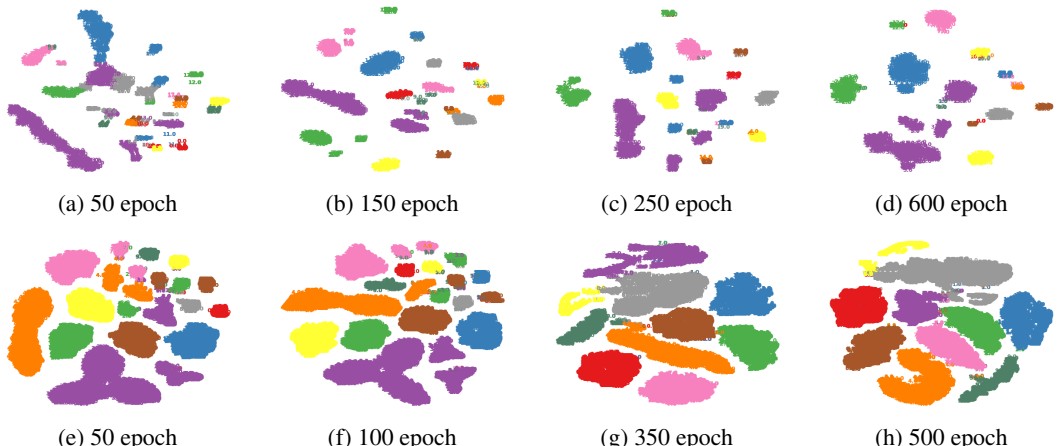

| (a) 50 epoch | (b) 150 epoch | (c) 250 epoch | (d) 600 epoch |
|---|---|---|---|
| (e) 50 epoch | (f) 100 epoch | (g) 350 epoch | (h) 500 epoch |

Figure 3: t-SNE visualizations of clustering results on the Caltech101-20 and Noisy MNIST datasets with increasing training iteration. (a)-(d) show the results on the Caltech101-20 dataset. (e)-(h) show the results on the Noisy MNIST dataset.

Table 3: Performance of HSACC on Mfeat dataset with different numbers of views.

| Views Number | Training Time (s) | ACC | NMI | ARI |
|:---:|:---:|:---:|:---:|:---:|
| 2 | 52.80 | 74.96 | 74.58 | 59.35 |
| 3 | 98.64 | 77.68 | 78.11 | 62.17 |
| 4 | 138.70 | 85.91 | 85.55 | 77.11 |
| 5 | 186.99 | 88.64 | 88.14 | 81.09 |

## 5 Conclusion

In this paper, we propose a novel framework, HSACC, based on hierarchical semantic alignment and cooperative completion. By designing a dual-level semantic space and employing a joint optimization strategy, the proposed method maps the learned latent representations into high-dimensional semantic spaces to implicitly infer missing view information, thereby providing more complete representations for subsequent clustering tasks. Extensive experimental results demonstrate that HSACC consistently outperforms existing mainstream methods across multiple datasets, verifying its effectiveness and superiority. In future work, we plan to extend this framework to more complex multi-modal incomplete data scenarios, and further enhance its generalization ability and computational efficiency in real-world large-scale applications.

## 6 Acknowledgements

This work was supported by the Project of Philosophy and Social Science Research in Colleges and Universities in Jiangsu Province (2024SJYB0220), and the Scientific Research Innovation Project of Nanjing University of Finance and Economics (XKYC2202507).

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

# A Technical Appendices and Supplementary Material

## A.1 Visualization

As shown in Figure 5, we perform t-SNE visualizations of the learned common representations on the LandUse_21 and Hdigit and 100leaves datasets. Figures (a)–(d) show the clustering results of the LandUse_21 dataset at different training epochs, Figures (e)–(h) present the clustering results of the Hdigit dataset at different training epochs, while Figures (i)–(l) show the clustering results of the 100leaves dataset at different training epochs. It can be observed that as the number of training epochs increases, the learned common representations exhibit increasingly clearer clustering structures in the feature space. The boundaries between different categories become more distinct, and the intra-class sample distributions become more compact, demonstrating stronger discriminative ability and improved clustering performance.

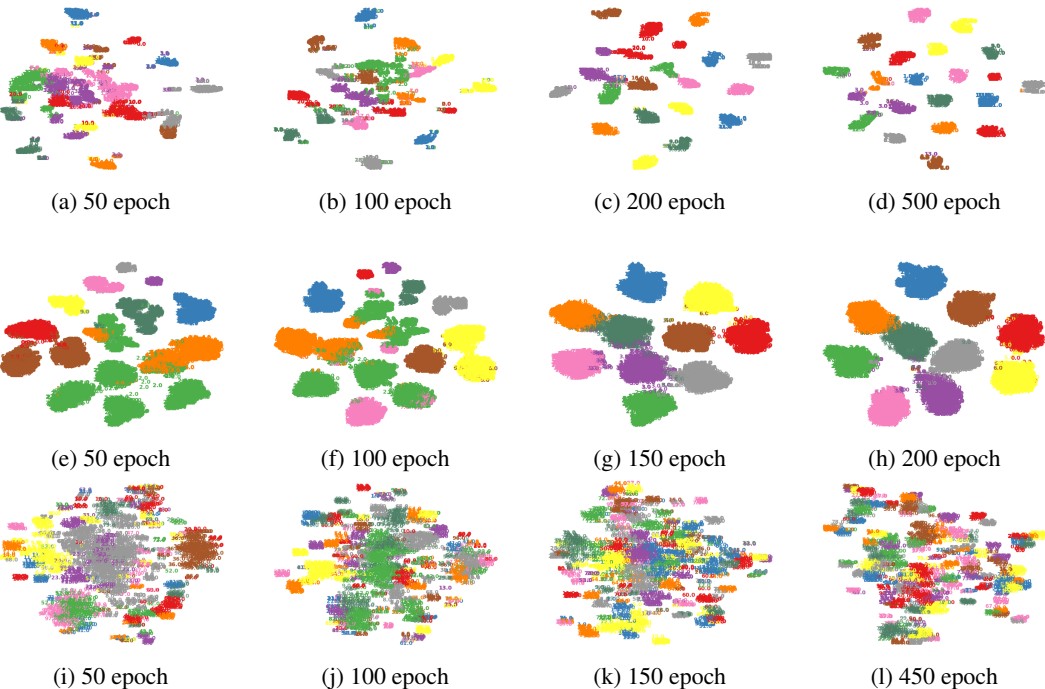

|                |                |                |                |
| :------------: | :------------: | :------------: | :------------: |
| (a) 50 epoch   | (b) 100 epoch  | (c) 200 epoch  | (d) 500 epoch  |
| (e) 50 epoch   | (f) 100 epoch  | (g) 150 epoch  | (h) 200 epoch  |
| (i) 50 epoch   | (j) 100 epoch  | (k) 150 epoch  | (l) 450 epoch  |

Figure 5: t-SNE visualizations of clustering results on the LandUse_21 and Hdigit and 100leaves datasets with increasing training iteration. (a)-(d) show the results on the LandUse_21 dataset. (e)-(h) show the results on the Hdigit dataset. (i)-(j) show the results on the 100leaves dataset.

