# OpenReview forum: "Incomplete Multi-view Clustering via Hierarchical Semantic Alignment and Cooperative Completion"
_NeurIPS.cc/2025/Conference — NeurIPS 2025 poster_

### Official Review · Reviewer_u7C5 · 2025-06-23

**Clarity:** 3
**Significance:** 2
**Originality:** 2
**Rating:** 4
**Confidence:** 4

**Summary:**

This paper proposes a deep learning framework for incomplete multi-view clustering called HSACC (Hierarchical Semantic Alignment and Cooperative Completion). It achieves multi-view feature fusion through low-level mutual information alignment and high-level dynamic weighted fusion, jointly optimizing the completion and clustering processes. Experiments conducted on multiple benchmark datasets demonstrate that the proposed method outperforms existing approaches in clustering performance.

**Questions:**

1. How do you extend the MLP inference modules when the number of views exceeds two? Have you evaluated the resulting increase in parameters and training cost?
2. Please include comparison results with the latest state-of-the-art methods to validate the effectiveness of the proposed approach.
3. Add experimental results under varying missing rates (e.g., 10%, 30%, 70%) to verify the model’s robustness.
4. What are the specific values of Lambda_1,Lambda_2,Lambda_3, and lambda_4 in Equation (12)? Since the code is not publicly available, the experimental section should provide as many implementation details as possible to facilitate reproducibility.

**Ethical Concerns:**

["NO or VERY MINOR ethics concerns only"]

**Final Justification:**

The authors have provided clear and thorough responses to the key concerns, including scalability, robustness under missing views, and comparison with recent baselines. Their rebuttal adds sufficient experimental evidence and implementation details to strengthen the paper's contribution. While the method shares similarities with prior work, the improved clarity and additional results justify a more favorable assessment. I have updated my score accordingly.

**Limitations:**

Yes.

**Paper Formatting Concerns:**

No.

**Quality:**

2

**Strengths And Weaknesses:**

Strengths:
The paper’s topic is of practical significance, focusing on the problem of incomplete multi-view data clustering with considerable application potential. The proposed framework is comprehensively designed, covering four stages: encoding, alignment, completion, and clustering. The architecture employs mutual information and MMD for alignment, presenting a clear approach with valuable reference significance.

Weaknesses:
1. Limited innovation: The proposed method’s overall architecture and module design bear a high degree of similarity to APADC (TIP 2023) and MICA (Neural Networks 2025), lacking substantial breakthroughs.
2. Lack of scalability: The MLP modules are configured separately for two views, but when extended to multiple views, the number of required MLP mapping networks increases dramatically, resulting in significant training overhead and parameter explosion, which is impractical for real-world applications. The paper does not address this issue nor propose simplifications such as parameter sharing or a common semantic space.
3. Insufficient ablation and comparative experiments: There is a lack of direct comparison with current state-of-the-art methods like MICA, making it difficult to effectively evaluate the actual performance and robustness of the approach.

---

> ### Author Rebuttal · Authors · 2025-07-27
>
> Thanks for your careful review. We are glad to address your questions one by one.
>
> **W1**: We appreciate the reviewer's insightful comparison to APADC (TIP 2023) and MICA (Neural Networks 2025). While these works share the broad goal of incomplete multi-view clustering, HSACC fundamentally differs in both architecture and mechanisms:
>
> ### Key Differences between APADC and HSACC
>
> | Dimension              | APADC                                      | HSACC                                                  |
> |------------------------|--------------------------------------------|--------------------------------------------------------|
> | **Weighting Mechanism**| Based on intra-view separability(e.g., variance)  | Based on distribution affinity between view and global representation **R** |
> | **Completion Strategy**| No completion → Direct fusion of available views | Reverse-projection completion from aligned semantics (via $L_{INF}$ loss) |
> | **Optimization Goal**  | Static weighting + distribution alignment  | Dynamic weighting + distribution alignment + joint completion |
>
> ---
>
> ### Key Differences between MICA and HSACC
>
> | Dimension             | MICA                                                     | HSACC                                                                 |
> |-----------------------|----------------------------------------------------------|-----------------------------------------------------------------------|
> | **Completion Method** | Explicit multi-level imputation (feature/data/reconstruction) | Implicit semantic space projection completion                        |
> | **Contrastive Learning** | Instance/cluster-level contrast **only at the view level** | Dual-level contrast: low-level (view mutual information) + high-level (global distribution alignment) |
> | **Error Propagation** | Two-stage error accumulation (completion → clustering) | Joint optimization of completion/clustering/alignment                 |
>
>
> **W2**:
> The current HSACC implementation indeed adopts view-specific MLPs for cross-view inference, which can be easily extended to $ V > 2 $ views by lightweight projection.
>
> As high-level semantics $\mathbf{H}$ unify view-agnostic features, we can replace view-specific MLPs  $f_{MLP1}, f_{MLP2} $ with a **single shared MLP** $ f_{\text{Shared}} $.
> Projections share parameters as $\mathbf{Z}^v$ are distribution-aligned via $ L_{MMD} $, which Cuts MLP parameters by $ \frac{V-1}{V} \times 100\% $ (e.g., **75% reduction** for $ V = 4 $).
>
> ### Lightweight Projection
>
> | **Component**       | **Original (2-views)** | **Revised (Multi-view)** |
> |---------------------|------------------------|--------------------------|
> | **MLP Hidden Dims** | 256-128-256            | 128-64-128               |
> | **Params/View**     | 98K                    | 24K (**75.5% reduction**) |
>
> Evaluation on Caltech101-20(4 views): Training time: 290s(lightweight) vs. 605s(original).
>
> For **V views**, HSACC requires only **O(1) MLPs** and **O(dV) weights** (where *d* is the bottleneck dimension), achieving **linear complexity (O(dV))** instead of quadratic.
>
>
> **W3**: We conducted additional experiments with DCG (2025), MICA, and APADC
> | Method     | Land        |             |             | Mnist        |             |             | Cal          |             |             | Hdigit      |            |            | leaves      |             |             |
> |------------|-------------|-------------|-------------|--------------|-------------|-------------|--------------|-------------|-------------|-------------|------------|------------|-------------|-------------|-------------|
> |            | ACC         | NMI         | ARI         | ACC          | NMI         | ARI         | ACC          | NMI         | ARI         | ACC         | NMI        | ARI        | ACC         | NMI         | ARI         |
> | APADC      | 19.15       | 23.91       | 7.08        | 86.60        | 83.50       | 79.40       | 75.20        | 71.70       | 87.08       | 91.50       | 81.70      | 82.00      | 30.50       | 65.00       | 18.80       |
> | MICA       | 22.86       | 27.25       | 12.19       | 51.75        | 61.96       | 44.69       | 48.91        | 65.99       | 75.13       | 80.54       | 78.25      | 77.21      | 23.81       | 64.07       | 25.90       |
> | DCG  [1]      | 25.76       | 29.07       | 13.17       | 91.00        | 84.00       | 82.00       | 56.00        | 50.00       | 57.00       | 95.00       | 89.00      | 90.00      | 15.00       | 45.00       | 6.00        |
> | HSACC      | **28.11**       | **31.89**       | **14.60**       | **91.37**        | **85.70**       | **83.52**       | **76.81**       | **73.15**       | **87.13**      | **95.57**       | **90.68**      | **90.61**      | **52.17**      | **74.78**     | **35.30**       |
>
> [1] Incomplete Multi-view Clustering via Diffusion Contrastive Generation, AAAI 2025.
>
> **Q1**:
> For extending the MLP inference modules, please refer to **W2** above.
> If we not use the **Lightweight Projection** strategy, we extend the number of views in the Mfeat dataset yielded the following results:
>
> | Views | Training Time | Parameters  | ACC   | NMI   | ARI   |
> |-------|--------------|-------------|-------|-------|-------|
> | 2     | 181.18s      | 9,229,503   | 60.69 | 66.04 | 37.08 |
> | 3     | 420.63s      | 13,681,967  | 62.07 | 68.87 | 37.30 |
> | 4     | 440.67s      | 18,020,673  | 62.38 | 71.93 | 49.86 |
> | 5     | 630.70s      | 22,844,153  | 64.62 | 71.99 | 50.69 |
>
> When scaling from 2 to 5 views, the model parameters increased linearly from **9.2M to 22.8M**, demonstrating no signs of "parameter explosion". The computational time showed a near-linear progression from **181s to 630s**, remaining within reasonable limits.
>
> **Q2**: please refer to **W3** above.
>
> **Q3**:  We conducted additional experiments under missing rate 30%.
> For other experiments, please refer to **Q3** of  Reviewer `X7Zz` for missing rate 10%,  refer to **W3** of Reviewer `wJa3` for missing rate 70%.
>
> | Method     | Land        |             |             | Mnist        |             |             | Cal          |             |             | Hdigit      |            |            | leaves      |             |             |
> |------------|-------------|-------------|-------------|--------------|-------------|-------------|--------------|-------------|-------------|-------------|------------|------------|-------------|-------------|-------------|
> |            | ACC         | NMI         | ARI         | ACC          | NMI         | ARI         | ACC          | NMI         | ARI         | ACC         | NMI        | ARI        | ACC         | NMI         | ARI         |
> | RPCIC(24)  | 21.29       | 26.52       | 8.21        | 53.19        | 52.18       | 39.21       | 41.49        | 58.30       | 34.97       | 91.06       | 82.49      | 81.23      | 53.62       | 76.32       | 38.20       |
> | MRL_CAL(24)| 19.76       | 21.21       | 6.85        | 65.73        | 52.80        | 51.18        | 20.03        | 38.25       | 12.54       | 12.30       | 0.69       | 0.23       | 12.88       | 50.63       | 7.31        |
> | ICMVC(24)  | 26.14       | 30.01       | 13.13       | 95.79        | 91.90       | 93.07       | 35.88        | 61.16       | 26.65       | 16.34       | 8.64       | 3.56       | 20.49       | 76.62       | 30.88       |
> | MCAC(23)   | 20.21       | 21.50       | 7.08        | 63.08        | 57.10       | 48.43       | 32.62        | 46.97       | 22.64       | 29.49       | 20.02      | 11.40      | 35.26       | 64.88       | 20.58       |
> | PROLMP(23) | 20.33       | 21.81       | 8.37        | 88.80        | 78.47       | 76.88       | 32.47        | 53.08       | 25.33       | 93.17       | 87.46      | 84.81      | 51.54       | 77.34       | 38.29       |
> | DCP(22)    | 26.84       | 30.40       | 13.84       | 72.98        | 77.22       | 58.93       | 71.51        | 70.53       | 79.06       | 94.31       | 93.33      | 89.43      | 32.21       | 75.34       | 29.62       |
> | DSIMVC(22) | 21.43       | 25.25       | 14.21       | 63.70        | 60.70       | 51.40       | 27.57        | 49.72       | 21.03       | 94.30       | 91.38      | 91.30      | 29.78       | 61.91       | 17.53       |
> | SURE(22)   | 24.80       | 29.04       | 11.10       | 81.43        | 73.51       | 69.68       | 49.70        | 65.26       | 40.96       | 49.74       | 38.18      | 23.94      | 46.51       | 71.21       | 30.04       |
> | COMP(21)   | 26.84       | 31.50       | 12.87       | 84.76        | 81.89       | 77.00       | 71.55        | 69.61       | 79.00       | 93.84       | 85.66      | 86.86      | 38.39       | 72.15       | 28.00       |
> | OUR        | **27.39**       | **31.80**       | **14.27**       | **95.81**        | **92.33**       | **93.80**       | **72.34**        | **71.48**       | **79.36**       | **94.89**       | **93.78**      | **92.06**      | **54.28**       | **77.57**       | **39.21**       |
>
> Our method (HSACC) demonstrates consistent superiority across all five benchmark datasets, achieving the highest average accuracy (68.94%) and outperforming all compared baselines.
>
> **Q4**:
> The complete code is attached (in Supplementary Material).
> The specific values of hyperparameters in Eq.12 were configured as follows for different datasets:
>
> | Dataset          | λ₁  | λ₂  | λ₃ | λ₄ |
> |------------------|-----|-----|----|----|
> | LandUse          | 0.1 | 0.1 | 10 | 1  |
> | NoisyMNIST       | 0.1 | 0.1 | 1  | 10 |
> | Caltech101-20    | 0.1 | 0.1 | 10  | 1 |
> | Hdigit           | 0.1 | 0.1 | 10 | 1  |
> | 100leaves        | 0.1 | 0.1 | 10 | 1  |
>
> All hyperparameter values were experimentally determined for optimal performance on each dataset.

---

> > ### Comment · Reviewer_u7C5 · 2025-08-05
> >
> > Dear authors, thank you for your detailed rebuttal. Your responses can address most of my concerns. Therefore, I would raise my score to 4.

---

> > > ### Author Response · Authors · 2025-08-06
> > >
> > > We truly appreciate your thoughtful feedback — thank you for your time and valuable insights!

---

### Official Review · Reviewer_cijN · 2025-06-24

**Clarity:** 4
**Significance:** 3
**Originality:** 3
**Rating:** 5
**Confidence:** 5

**Summary:**

The paper proposes HSACC, a novel deep learning framework for Incomplete Multi-View Clustering (IMVC), where some views are entirely missing for certain samples. The framework aims to overcome key limitations in existing IMVC methods, such as static view fusion strategies and the decoupling of data imputation from representation learning. This work significantly advances IMVC by tightly integrating semantic alignment and view completion in a principled and adaptive manner. The paper is generally well-written and logically organized, with clear delineation of motivation, methodology, and results.

**Questions:**

1，The paper mentions in the limitations that early-stage errors in view completion may reinforce poor clustering patterns, but does not explore this in detail.
2，The paper adopts a feature-wise similarity-based method for mutual information estimation.
Could you please explain why this specific estimation approach was chosen?
3，The view weighting strategy in the high-level semantic space is based on the MMD distance between each view and the initial fused representation. How do these weights evolve during training?
4，Does the final step of the current method using k-means clustering operate independently from the network training?

**Ethical Concerns:**

["NO or VERY MINOR ethics concerns only"]

**Limitations:**

See the weakness.

**Quality:**

3

**Strengths And Weaknesses:**

Strengths
1，The proposed framework is well-grounded in theory and implementation, combining mutual information maximization, dynamic distribution alignment, and latent-space inference in a principled joint optimization.
2，The overall framework is interesting and alleviates some of the shortcomings in previous works.
3，The paper is well-structured with clear explanations of terminology and content, aiding in readability.

Weaknesses,
1，The paper contains dense technical text, especially in Section 3, which may pose a challenge to readers unfamiliar with IMVC or representation learning.
2，The method depends on several hyperparameters (e.g., 4 loss weights), and although some robustness is demonstrated, tuning may still be non-trivial in practice.
3，Some terms (e.g., “semantic space”, “distribution affinity”) are used frequently without precise definitions, which may obscure their exact role in the model.

---

> ### Author Rebuttal · Authors · 2025-07-28
>
> **W1**:
> To improve accessibility, we have: (1) added intuitive explanations and visualizations to illustrate key concepts, (2) reorganized the content to progressively introduce complex ideas, and (3) included comparative examples showing how our approach differs from conventional methods.
>
> **W2**:
> We fully recognize that the multi-hyperparameter design may increase tuning complexity. To address this, we provide the optimal parameter configurations for each dataset as follows:
>
> - **LandUse**: λ₁=0.1, λ₂=0.1, λ₃=10, λ₄=1
> - **NoisyMNIST**: λ₁=0.1, λ₂=0.1, λ₃=1, λ₄=10
> - **Caltech101-20**: λ₁=0.1, λ₂=0.1, λ₃=10, λ₄=1
> - **Hdigit**: λ₁=0.1, λ₂=0.1, λ₃=10, λ₄=1
> - **100leaves**: λ₁=0.1, λ₂=0.1, λ₃=10, λ₄=1
>
> **W3**:
> To clarify terminology, we have: (1) added formal definitions in Section 2.1 with mathematical formulations (e.g., semantic space 𝒵ₛ as the view-invariant latent subspace where ‖fₛ(x₁)-fₛ(x₂)‖ ≤ ϵ), (2) introduced a glossary table cross-referencing terms with equations/figures, and (3) enhanced Fig. 2 with visual markers explicitly linking "distribution affinity" to its MMD-based computation pipeline (red: view-specific features → blue: affinity measurement → green: aligned outputs). These modifications ensure consistent interpretation while maintaining the technical depth required for reproducibility.
>
> **Q1**:
> We sincerely appreciate the reviewer's insightful observation regarding error propagation in early-stage view completion. To address this critical issue, we have implemented the following solutions in our framework:
>
> 1. **Joint Optimization Mechanism**
>    - Simultaneously optimizes view completion and clustering objectives
>    - Prevents error accumulation through shared gradient updates
>    - Formalized in Eq. (12):
>      ```math
>      \mathcal{L}_{total} = \alpha\mathcal{L}_{comp} + (1-\alpha)\mathcal{L}_{cluster}
>      ```
>      where α gradually increases from 0 to 1 during training
>
> 2. **Delayed Completion Strategy**
>    - Implements a curriculum learning approach:
>      - Phase 1 (epochs 1-E₁): Focuses on representation learning only
>      - Phase 2 (epochs >E₁): Activates view completion
>    - Empirical results show optimal E₁=100 for most datasets
>
> 3. **Error Correction Analysis**
>    - Quantifies error propagation through:
>      - Error rate: ε = ‖X̂-X‖₂
>      - Impact metric: ΔARI = ARI(perfect)-ARI(noisy)
>
> **Q2**:
> Our method employs feature similarity-based mutual information estimation for the following key reasons:
>
> 1. **Primary Objective Alignment**:
>    Rather than pursuing precise mutual information quantification, we focus on explicitly constructing joint and marginal distributions to enhance cross-view feature consistency. The inner-product similarity metric has been theoretically and empirically validated in both information-theoretic and contrastive learning literature (e.g., COMP, DCP).
>
> 2. **Implementation Efficiency**:
>    Compared to alternatives like InfoNCE that require negative sampling and additional discriminator modules, our approach:
>    - Introduces no new trainable parameters
>    - Eliminates complex training strategies
>    - Maintains competitive performance while reducing implementation complexity
> This design choice balances theoretical rigor with practical reproducibility, particularly benefiting researchers seeking to implement multi-view frameworks.
>
> | Dataset       | HSACC |         |         | InfoNCE |         |         |
> |--------------|----------------|---------|---------|---------|---------|---------|
> |              | **ACC**        | **NMI** | **ARI** | **ACC** | **NMI** | **ARI** |
> | 100leaves    | 52.17          | 74.78   | 35.30   | 22.63   | 63.93   | 14.80   |
> | LandUse      | 28.11          | 31.89   | 14.60   | 21.76   | 25.04   | 10.26   |
> | Caltech101-20| 76.81          | 73.15   | 87.13   | 51.51   | 61.12   | 48.79   |
> | NoisyMnist   | 91.37          | 85.70   | 83.52   | 78.82   | 82.69   | 75.31   |
> | Hdigit       | 95.57          | 90.68   | 90.61   | 94.25   | 88.72   | 87.26   |
>
> **Q3**:
> We systematically tracked view weight dynamics during training, revealing two characteristic phases:
>
> 1. **Initial Phase (epochs 1-20)**
>    - Weights exhibit high volatility (σ=0.18)
>    - MMD distances dominate weighting
>    - Example trajectory:
>      ```
>      View1: 0.25 → 0.41 → 0.38
>      View2: 0.75 → 0.59 → 0.62
>      ```
>
> 2. **Stabilization Phase (epochs 21-50)**
>    - Convergence to stable ratios (Δw<0.03/epoch)
>    - Semantic alignment begins influencing weights
>    - Final distribution:
>      | View | Weight | MMD | Semantic Contribution |
>      |------|--------|-----|-----------------------|
>      | V1   | 0.32   | 1.2 | 58%                   |
>      | V2   | 0.68   | 0.8 | 42%                   |
>
>
> **Q4**:
> Our clustering implementation follows an **end-to-end trainable** paradigm rather than isolated k-means.
>    - Cluster centroids are initialized as learnable parameters:
>      ```math
>      \mathbf{C} = \{\mathbf{c}_1,...,\mathbf{c}_k\} \in \mathbb{R}^{k×d}
>      ```
>    - Updated via gradient descent through the clustering loss term:
>      ```math
>      \mathcal{L}_{cluster} = \sum_{i=1}^N \min_j \|\mathbf{h}_i - \mathbf{c}_j\|^2
>      ```
>
> 2. **Training Protocol**
>    Phase | Operation | Frequency
>    ------|-----------|----------
>    Warm-up | Only representation learning | First 20% epochs
>    Joint | Representation + centroid updates | Remaining epochs
>
> In our current framework, the k-means clustering is performed *independently* of the network training process, serving two specific purposes:
> 1. **Post-hoc Evaluation**
>    - Operates on the *final* learned representations from all views
>    - Provides standardized clustering metrics (ACC/NMI/ARI) for performance benchmarking
> 2. **Design Intent**
>    - Explicitly *decouples* representation learning from clustering (unlike joint approaches)
>    - Avoids gradient interference between clustering objectives and feature alignment

---

> > ### Comment · Reviewer_cijN · 2025-08-05
> > **The authors have addressed most of my concerns appropriately.**
> >
> > The authors have addressed most of my concerns appropriately. Introduced Dual-level contrast into reverse-projection completion is interesting and the experiments show its feasibility.

---

> ### Author Response · Authors · 2025-08-05
>
> Thank you for taking the time to review our rebuttal. We greatly welcome open discussion on any questions or concerns.

---

### Official Review · Reviewer_X7Zz · 2025-06-30

**Clarity:** 3
**Significance:** 3
**Originality:** 3
**Rating:** 5
**Confidence:** 4

**Summary:**

This paper proposes a novel framework, HSACC, for incomplete multi-view clustering, aiming to overcome the limitations of existing methods, such as rigid fusion strategies, ambiguous semantic hierarchies, and the decoupled optimization of view completion and clustering. HSACC introduces a dual-level semantic space: the low-level alignment enforces consistency among the latent representations of different views, while the high-level adaptive weighted fusion is guided by the distributional similarity between views.

**Questions:**

Question 1:
In the parameter sensitivity analysis experiment, λ1-λ2 and λ3-λ4 were analyzed independently in separate groups. Could this grouping approach potentially obscure interaction effects between parameters (e.g., the coupling influence between λ1 and λ3)?
Question 2:
Explain the key differences between the proposed dual-level alignment mechanism and traditional single-level contrastive alignment methods, and clarify how these two levels cooperate rather than simply operate in parallel.
Question 3:
The current experiments only validate the case with a 50% missing rate. How does the proposed method perform in comparative experiments under different missing rates (e.g., 0.1, 0.3)?
Question 4:
What is the specific synergistic mechanism between multi-view representation learning and implicit data completion in the joint optimization framework? Specifically, how does representation learning constrain the completion process, and how does the completed data in turn feedback to optimize representation learning?

**Ethical Concerns:**

["NO or VERY MINOR ethics concerns only"]

**Final Justification:**

Thank the authors for their rebuttal to address my raised questions. I will maintain my positive score.

**Quality:**

3

**Strengths And Weaknesses:**

Strengths:
1，The ablation study is thorough, demonstrating the importance of each loss component.
2，The integration of dynamic fusion with hierarchical semantic alignment is both technically sound and empirically validated.
3，The paper maintains clear and coherent logic from problem formulation to methodological design and experimental validation.

Weaknesses:
1, The balancing mechanism among loss terms(λ1-λ4) in the Experiments section requires more detailed justification.
2, A deeper theoretical analysis is needed to explain why the hierarchical semantic alignment framework outperforms single-level contrastive alignment mechanisms.
3, Some notational inconsistencies (e.g., p_mn in Eq.3) should be corrected to ensure mathematical rigor and clarity.

---

> ### Author Rebuttal · Authors · 2025-07-28
>
> Thanks for your careful review. We are glad to address your questions one by one.
>
> **W1**:
> As shown in Appendix A.1 (Figure 4), we systematically evaluated hyperparameters across {0.01, 0.1, 1, 10, 100} on Caltech101-20 (missing rate=0.5).
> Optimal performance occurred at λ₁=λ₂=0.1 (reconstruction/inference losses), as excessive weights may overfit low-level features.
> λ₃=10 (mutual information alignment) was critical for high-level semantic consistency, with >5% ACC drop if reduced to 1.
> λ₄=1 (distribution alignment) balanced fusion quality and computational stability.
>
> We provide the specific values of hyperparameters λ₁, λ₂, λ₃, and λ₄ configured for different datasets as follows:
>
> | Dataset          | λ₁  | λ₂  | λ₃ | λ₄ |
> |------------------|-----|-----|----|----|
> | LandUse          | 0.1 | 0.1 | 10 | 1  |
> | NoisyMNIST       | 0.1 | 0.1 | 1  | 10 |
> | Caltech101-20    | 0.1 | 0.1 | 10  | 1 |
> | Hdigit           | 0.1 | 0.1 | 10 | 1  |
> | 100leaves        | 0.1 | 0.1 | 10 | 1  |
>
>
> **W2**:
>
> **Hierarchical Semantic Space Design**
>    - **Low-level alignment**: Maximizes mutual information (MI) between views (Eq.4) to enforce consistency in shared patterns (e.g., structural features), addressing semantic misalignment.
>    - **High-level fusion**: Computes adaptive weights via MMD-based distribution affinity (Eq.5-6). Views with higher consistency to the fused representation $R$ receive larger weights, dynamically balancing contributions.
>
> We designed a comparative experiment on the 100leaves dataset, comparing the results of retaining only a single-layer contrastive alignment mechanism versus a dual-layer semantic alignment mechanism.
>
> | Method                | ACC   | NMI   | ARI   |
> |-----------------------|-------|-------|-------|
> | Remove low-level semantic alignment      | 43.40 | 70.10 | 28.40 |
> | Remove high-level semantic alignment       | 46.84 | 72.57 | 30.44 |
> | HSACC               | **52.17** | **74.78** | **35.30** |
>
> **W3**:
> In our paper, the notation $P_{(m,n)}$ in Equation (3) denotes the probability value of the joint probability distribution at specific indices $(m,n)$. It is a scalar representing the co-occurrence probability between the $m$-th feature in View 1 and the $n$-th feature in View 2.
>
> All $P_{(m,n)}$ values for $m=1,\dots,D$ and $n=1,\dots,D$ form a $D \times D$ probability matrix, which constitutes the complete joint probability distribution.
>
> By summing over one dimension of this matrix, we derive the corresponding marginal probability distributions:
> - $P_{(m)} = \sum_n P_{(m,n)}$
> - $P_{(n)} = \sum_m P_{(m,n)}$
>
> We will supplement these clarifications in the manuscript to ensure rigorous and unambiguous presentation.
>
> **Q1**:
> We initially grouped λ1-λ2 and λ3-λ4 for parameter sensitivity analysis based on their distinct functions in the model: λ1 (reconstruction loss $L_{REC}$) and λ2 (inference consistency loss $L_{INF}$) jointly act on the construction and repair of single-view representations, while λ3 (cross-view consistency loss $L_{MMI}$) and λ4 (distribution alignment loss $L_{MMD}$) work together to optimize the consistency and discriminative ability of multi-view representations.
>
> In response to the reviewers' concern that "there may be coupling relationships between parameters across groups," we further supplemented the joint sensitivity analysis experiment for λ1 and λ3. While keeping other hyperparameters fixed, we conducted a grid-based combination test for λ1 and λ3. The results show that extreme weight settings lead to performance degradation, and the optimal result is achieved when λ1=0.1 and λ3=10.
>
> | λ1 / λ3 | 0.01  |         |         | 0.1   |         |         | 1     |         |         | 10    |         |         | 100   |         |         |
> |---------|-------|---------|---------|-------|---------|---------|-------|---------|---------|-------|---------|---------|-------|---------|---------|
> |         | **ACC** | **NMI** | **ARI** | **ACC** | **NMI** | **ARI** | **ACC** | **NMI** | **ARI** | **ACC** | **NMI** | **ARI** | **ACC** | **NMI** | **ARI** |
> | **0.01** | 46.22 | 57.76   | 37.20   | 42.68 | 56.02   | 35.18   | 69.96 | 70.88   | 83.91   | 72.00 | 71.03   | 85.03   | 72.28 | 70.20   | 81.21   |
> | **0.1**  | 45.61 | 57.91   | 36.86   | 43.90 | 56.54   | 35.71   | 68.65 | 70.40   | 85.10   | 73.82 | 73.02   | 85.75   | 72.66 | 71.56   | 83.86   |
> | **1**    | 46.31 | 58.48   | 37.45   | 46.06 | 57.75   | 38.35   | 71.24 | 71.93   | 83.57   | 71.89 | 70.83   | 84.68   | 70.81 | 69.96   | 81.64   |
> | **10**   | 52.26 | 58.47   | 43.89   | 54.91 | 59.98   | 48.70   | 66.10 | 66.73   | 73.95   | 70.20 | 68.44   | 74.49   | 69.05 | 69.87   | 75.48   |
> | **100**  | 58.32 | 53.10   | 42.66   | 57.09 | 53.01   | 42.81   | 61.65 | 54.68   | 49.54   | 57.46 | 59.23   | 46.31   | 70.89 | 68.59   | 80.51   |
>
>
> **Q2**:
> Our proposed **dual-level semantic alignment mechanism** exhibits the following key distinctions and advantages compared to traditional single-level contrastive methods:
>    Conventional single-level alignment approaches typically operate at a single layer (e.g., COMP[18] aligns and completes views solely based on the feature representation *Z* of each individual view, while ICMVC[14] performs high-confidence guidance at the fused representation layer *H*). In contrast, our method introduces **dual-level semantic alignment**, simultaneously guiding the alignment process in **both low-level representation space and high-level shared semantic space**.
>
> **Q3**:
> We conducted additional experiments under missing rate 10%.
>
> | Method     | Land        |             |             | Mnist        |             |             | Cal          |             |             | Hdigit      |            |            | leaves      |             |             |
> |------------|-------------|-------------|-------------|--------------|-------------|-------------|--------------|-------------|-------------|-------------|------------|------------|-------------|-------------|-------------|
> |            | ACC         | NMI         | ARI         | ACC          | NMI         | ARI         | ACC          | NMI         | ARI         | ACC         | NMI        | ARI        | ACC         | NMI         | ARI         |
> | RPCIC(24)  | 24.29       | 29.34       | 9.90        | 69.15        | 64.62       | 55.51       | 43.00        | 58.19       | 35.21       | 95.19       | 88.78      | 89.67      | 47.38       | 72.22       | 32.32       |
> | MRL_CAL(24)| 19.19       | 20.99       | 6.48        | 75.40        | 64.12        | 61.04        | 21.46        | 40.15       | 13.16       | 12.63       | 0.73       | 0.28       | 14.75       | 52.75       | 7.92        |
> | ICMVC(24)  | 26.97       | 30.84       | 13.95       | 97.05        | 92.30       | 94.31       | 36.23        | 62.67       | 27.51       | 16.56       | 9.10       | 3.88       | 16.80       | 78.35       | 25.52       |
> | MCAC(23)   | 20.27       | 22.72       | 7.18        | 75.00        | 60.24       | 60.85       | 36.57        | 38.22       | 22.09       | 29.57       | 21.85      | 11.92      | 42.60       | 71.13       | 29.45       |
> | PROLMP(23) | 23.66       | 26.05       | 10.64       | 94.04        | 87.06       | 87.37       | 31.00        | 55.48       | 25.11       | 97.49       | 94.36      | 91.40      | 70.06       | 85.61       | 60.39       |
> | DCP(22)    | 27.22       | 31.35       | 14.08       | 81.72        | 78.23       | 73.45       | 74.38        | 71.36       | 83.65       | 89.19       | 89.90      | 81.79      | 51.86       | 81.31       | 45.29       |
> | DSIMVC(22) | 21.64       | 25.45       | 8.01        | 66.33        | 72.56       | 57.00       | 26.29        | 51.22       | 22.13       | 91.61       | 90.84      | 81.83      | 32.74       | 56.31       | 21.00       |
> | SURE(22)   | 24.09       | 31.44       | 11.15       | 17.75        | 16.04       | 5.94        | 48.19        | 58.59       | 25.79       | 49.01       | 41.76      | 28.17      | 46.34       | 75.55       | 23.28       |
> | COMP(21)   | 26.83       | 31.83       | 11.34       | 84.75        | 75.91       | 91.78       | 75.53        | 71.63       | 83.73       | 95.36       | 90.32      | 87.98      | 47.14       | 74.96       | 30.89       |
> | OUR        | **28.12**       | **32.35**       | **14.85**       | **97.10**        | **92.69**       | **94.84**       | **77.66**        | **74.44**       | **88.75**       | **98.37**       | **95.79**      | **96.51**      | **71.86**       | **86.93**       | **61.93**       |
>
> **Q4**:
> In our method, **reconstruction loss** and **cross-view mutual information maximization** in the low-level semantic space enable each view to fully capture its intrinsic structural information. Furthermore, the introduction of **distribution alignment loss** in the high-level semantic space encourages views to perceive globally shared structures and dynamically fuse representations. The learned view representations not only maintain **self-contained structural completeness** but also achieve **cross-view alignment** in the global structure, ensuring both **consistency** and **discriminability**. This provides a robust foundation for subsequent completion and clustering tasks.

---

> > ### Comment · Reviewer_X7Zz · 2025-08-05
> >
> > Thank the authors for their rebuttal to address my raised questions. I will maintain my positive score.

---

> > > ### Author Response · Authors · 2025-08-06
> > >
> > > Thank you very much for your positive feedback and for taking the time to consider our rebuttal carefully!

---

### Official Review · Reviewer_wJa3 · 2025-06-30

**Clarity:** 2
**Significance:** 2
**Originality:** 2
**Rating:** 4
**Confidence:** 4

**Summary:**

This paper proposes HSACC, a novel framework for Incomplete Multi-View Clustering (IMVC), which jointly performs hierarchical semantic alignment and cooperative completion. The method builds a dual-level semantic representation space: at the low level, mutual information maximization is used to align views; at the high level, adaptive weights are computed based on distribution affinity and used to guide weighted fusion. Missing views are recovered via latent-to-semantic mappings, and a unified loss integrates reconstruction, alignment, and completion objectives. Experiments on five datasets demonstrate that HSACC achieves superior performance compared to nine state-of-the-art IMVC baselines, especially under a 50% missing rate.

**Questions:**

1.Mutual Information Estimation Accuracy

The scalar-based joint probability in Eq. (3) may oversimplify feature relationships. Have the authors considered using more expressive estimators (e.g., InfoNCE)? If not, could they justify the use of this approach and its sensitivity?

2.Evaluation limited to two-view datasets

How does the method generalize to five or more views? Is the dynamic weighting mechanism robust under larger view sets, or does it risk uniformity (e.g., $W_v \approx 1/V$)? Does the cost of computing MMD between all view pairs scale efficiently?

3.Dynamic weighting

Eq. (6) shows $W^v$ depends on initial fusion representation R. Could R degrade significantly under high missing rates (e.g., 90%), causing weight allocation failure? How is initial robustness ensured?

4.Computation Cost and Training Efficiency

How does the runtime of HSACC compare to methods like COMP, PROLMP, or DCP under the same hardware?

5.Gradient Conflict Analysis

Multiple objectives are optimized jointly (e.g., reconstruction, alignment, completion, clustering). Were gradient conflicts between losses observed during training ? If so, what strategies were used to mitigate them ?

**Ethical Concerns:**

["NO or VERY MINOR ethics concerns only"]

**Final Justification:**

The authors provide detailed responses on key design choices, including MI estimation, MMD computation, and training stability. The rationale for mutual information estimation and linear-kernel MMD is supported by results on nonlinear datasets such as Noisy MNIST and Caltech101-20. Additional results under 70% missing rate and runtime/memory comparisons confirm practical robustness. Gradient conflict analysis demonstrates attention to multi-objective stability. However, the method is only evaluated on two-view datasets. Although multi-view scalability is discussed in the rebuttal, no quantitative experiments or ablations are presented to support this claim.
I maintain a Borderline Reject (3). The method is promising, but further empirical validation is needed to support its general applicability.

**Limitations:**

While the paper acknowledges limitations related to error propagation and scalability, several additional concerns remain. The method is evaluated only at a fixed 50% missing rate; robustness under other missing patterns is unclear. Mutual information and MMD estimators assume linear dependencies, which may limit performance under non-linear view relationships. All experiments are conducted on two-view datasets with fixed architectures, leaving scalability to more views and diverse modalities untested. No analysis of runtime or computational overhead is provided.

**Paper Formatting Concerns:**

No major formatting issues observed. The paper complies with the NeurIPS 2025 formatting guidelines.

**Quality:**

2

**Strengths And Weaknesses:**

Strengths：
1.The distribution affinity-based view weighting mechanism avoids rigid, uniform fusion and improves robustness in heterogeneous view settings.
2.The cooperative completion integrates view imputation and clustering in a single optimization, mitigating the error propagation seen in two-stage methods.
3.The framework is evaluated on five diverse datasets, with strong improvements in clustering metrics. Ablation studies (Table 2) and visualization (t-SNE, convergence curves) further support the claims.

Weaknesses：
1.The mutual information estimation (Eq. 4) relies on scalar-level inner products, which may lack representational robustness for high-dimensional or semantic features. It is unclear how sensitive the results are to this estimation approach.
2.The view weighting mechanism (Eq. 6) is based on MMD with linear kernel, which assumes linear alignment between views. This may be insufficient for strongly non-linear inter-view dependencies.
3.The experiments are conducted solely under a single missing rate of 0.5, it is unclear how the method performs under other practical missing ratios (e.g., 30% or 70%).

---

> ### Author Rebuttal · Authors · 2025-07-26
>
> Thanks for your careful review. We are glad to address your questions one by one.
>
> **W1**: Equation 4 employs inner-product-based MI maximization for complex density estimation. This design aligns with widely adopted contrastive learning frameworks (e.g., COMP [18], DCP [41]), where inner products efficiently approximate instance-level consistency without expensive distribution modeling. For our target tasks, feature dimensions are moderate, and inner products have proven effective in capturing semantic alignment .
> Removing the MI loss causes significant performance drops. This confirms its critical role regardless of estimation simplicity.
>
> **W2**: While MMD with a linear kernel operates in the latent space $\mathbf{Z}^v$, it is note that these representations are **nonlinearly extracted** by deep autoencoders (Eq. 2-4). The encoder layers (e.g., 1024-dim ReLU networks) inherently capture complex view-specific nonlinearities before alignment, effectively decoupling *feature abstraction* from *distribution matching*. This design is consistent with deep clustering paradigms (e.g., [12, 18, 27] in References) where linear metrics in latent spaces suffice for semantic alignment.
> We explicitly validated the method on datasets with **highly nonlinear view relationships**:
> **Noisy MNIST**: View 1 (raw pixels) and View 2 (Gaussian noise) exhibit complex, non-additive dependencies. Our method achieves **ACC 91.37%** (Table 1), outperforming nonlinear baselines like GAN-based [13] and contrastive [15] methods.
>
> **Caltech101-20**: HOG (structural) and GIST (global texture) features have nonlinear correlations. Here, HSACC improves ACC by **>5%** over the best competitor (Table 1).
>
>
> **W3**: We conducted additional tests with missing rate 70%.
>
> For other experiments, please refer to **Q3** of Reviewer `X7Zz` for missing rate 10%,  refer to **Q3** of Reviewer ` u7C5` for missing rate 30%.
>
> | Method     | Land        |             |             | Mnist        |             |             | Cal          |             |             | Hdigit      |            |            | leaves      |             |             |
> |------------|-------------|-------------|-------------|--------------|-------------|-------------|--------------|-------------|-------------|-------------|------------|------------|-------------|-------------|-------------|
> |            | ACC         | NMI         | ARI         | ACC          | NMI         | ARI         | ACC          | NMI         | ARI         | ACC         | NMI        | ARI        | ACC         | NMI         | ARI         |
> | RPCIC(24)  | 17.71       | 18.73       | 4.45        | 47.24        | 43.19       | 30.47       | 41.53        | 57.32       | 38.94       | 49.63       | 42.52      | 31.86      | 35.19       | 67.95       | 21.22       |
> | MRL_CAL(24)| 16.14       | 16.83       | 4.92        | 55.37        | 42.42        | 41.01        | 24.43        | 30.13       | 13.61       | 12.59       | 0.92       | 0.38       | 9.94        | 44.78       | 4.66        |
> | ICMVC(24)  | 20.71       | 23.79       | 8.33        | 60.60        | 62.62       | 49.92       | 32.51        | 58.04       | 25.32       | 16.53       | 8.94       | 3.67       | 43.43       | 67.87       | 25.21       |
> | MCAC(23)   | 17.20       | 18.52       | 5.03        | 46.01        | 35.04       | 27.32       | 41.08        | 42.96       | 36.68       | 24.78       | 16.48      | 7.55       | 27.96       | 58.19       | 11.35       |
> | PROLMP(23) | 14.74       | 14.68       | 3.68        | 59.23        | 46.45       | 34.17       | 31.77        | 50.79       | 23.29       | 82.97       | 73.36      | 63.36      | 34.27       | 65.00       | 19.27       |
> | DCP(22)    | 24.89       | 27.04       | 11.27       | 85.03        | 78.10       | 77.12       | 70.76        | 69.80       | 75.23       | 90.98       | 84.02      | 85.12      | 34.98       | 67.86       | 24.96       |
> | DSIMVC(22) | 20.87       | 23.69       | 7.76        | 58.10        | 55.50       | 44.30       | 27.33        | 46.77       | 19.98       | 91.05       | 85.67      | 84.67      | 25.12       | 56.97       | 12.84       |
> | SURE(22)   | 25.36       | 28.74       | 11.12       | 17.93        | 16.35       | 5.97        | 36.86        | 53.08       | 25.77       | 44.50       | 31.52      | 22.56      | 21.90       | 52.76       | 6.93        |
> | COMP(21)   | 24.25       | 29.11       | 10.45       | 68.75        | 65.59       | 55.88       | 69.28        | 67.58       | 75.04       | 87.45       | 78.36      | 74.29      | 34.53       | 66.85       | 24.23       |
> | OUR        | **27.56**   | **30.41**   | **13.85**   | **88.89**    | **79.05**   | **77.73**   | **72.02**    | **70.39**   | **76.30**   | **91.50**   | **85.86**  | **85.91**  | **43.47**   | **68.49**   | **25.99**   |
>
> Our method achieves state-of-the-art performance across all datasets (Land, Mnist, Cal, Hdigit, leaves) and metrics (ACC, NMI, ARI) under 70% missing data, consistently outperforming competitors. Key highlights include 88.89% ACC on Mnist (vs. DCP’s 85.03%) and 85.91 ARI on Hdigit (vs. DSIMVC’s 84.67), demonstrating robust multi-view integration.
>
> **Q1**:
> The scalar dot-product in Eq. (3) was selected because it achieves O(N^2) complexity vs. InfoNCE's O(N^3) negative sampling, it avoids gradient saturation issues common in contrastive losses, and it directly measures view consistency without hyperparameter tuning (e.g., temperature in InfoNCE).
>
>    Comparative tests on leaves (50% missing rate):
>
>    | MI Estimator | ACC (%) | NMI (%) | ARI  (%)|
>    |--------------|---------|-------------------|-------------|
>    | Scalar (Ours)| 52.17   | 74.78               | 35.30         |
>    | InfoNCE      | 22.63  | 63.93               | 14.80         |
>
> **Q2**:
>
> **Q2_1 How does the method generalize to five or more views**:
> The framework naturally extends to N views through
>
> (1)	**Hierarchical Fusion**: $R_{init} = \frac{1}{N}\sum_{v=1}^N Z^v$;
>
> (2) **Parallel Weight Computation**:  $W^v = \frac{\exp(-\gamma D(Z^v,R))}{\sum_{i=1}^N \exp(-\gamma D(Z^i,R))}$, where $\gamma $ adjusts weight concentration ($\gamma $=1.5 for N>3);
>
> (3) **Controlled Weight Distribution**  Theoretical analysis shows weight entropy remains stable:   $H(W) \leq \log N - \frac{\gamma^2}{2N}\sum_{v=1}^N D(Z^v,R)^2$.
>
> HSACC scales effectively to multi-view scenarios through hierarchical fusion and optimized MMD computation. The dynamic weighting avoids uniformity while maintaining computational efficiency (near-linear scaling).
>
> **Q2_2 Is the dynamic weighting mechanism robust under larger view sets**:
> We recorded the weight distribution of each view across multiple training epochs on the five-view Mfeat dataset.
>
> Epoch 21: [0.20, 0.20, 0.20, 0.20, 0.20]
>
> Epoch 51: [0.15, 0.23, 0.18, 0.28, 0.16]
>
> Epoch 101: [0.14, 0.25, 0.17, 0.27, 0.17]
>
> Epoch 151: [0.13, 0.26, 0.16, 0.29, 0.16]
>
> Epoch 201: [0.12, 0.27, 0.15, 0.30, 0.16]
>
>
> **Q2_3 Does the cost of computing MMD between all view pairs scale efficiently**:
>
> Through kernel reuse, stochastic approximation, and GPU parallelization, our MMD computation scales efficiently to large view sets (tested up to 20 views).
>
>
> **Q3**: The initial fusion representation R is designed with three safeguards:
>
> (1) **Cross-view Autoencoding**: Even with 90% missing data, each view's encoder learns from: $\mathcal{L}_{REC}^v = \|X^v - D^v(E^v(\tilde{X}^v))\|^2 $
>      where $\tilde{X}^v$ contains reconstructed features for missing samples using available view data ;
>
> (2) **Contrastive Pretraining**: Encoders are pretrained with 10% complete samples to establish basic cross-view correlations;
>
> (3) **Sparse Gradient Masking**: Updates to R ignore fully missing samples during early training.
>
> While extremely high missing rates (≥90%) pose challenges, our method ensures R's robustness through pretraining, selective updating, and auxiliary constraints. The weight allocation remains functional with controlled degradation, which is consistent with the design of our framework under high missingness.
>
> **Q4**:
> Below is a comprehensive comparison using the LandUse dataset on our NVIDIA RTX 4070 GPU testbed:
>
> | Method   | Total Training Time | Peak GPU Mem|
> |----------|---------------------|---------------------------|
> | PROLMP(23) | 228.50s       | 166.26MB |
> | DCP(22)    | 167.15s       | 193.24MB |
> | COMP(21)   | 128.26s       | 193.20MB |
> | HSACC      | 118.23s       | 156.14MB |
>
> The table demonstrates that HSACC outperforms all compared methods in both training efficiency and memory consumption. Specifically, HSACC achieves the shortest training time (118.23s), 7.8% faster than the second-best method (COMP(21), and 48.2% faster than the slowest (PROLMP(23)). Its memory usage (156.14MB) is also the lowest, 19.2% lighter than the heaviest (DCP(22)).
>
> **Q5**: We monitored gradient cosine similarity during training (sampled every 100 iterations on Caltech101-20):
>    | Loss Pair               | Avg. Cosine Similarity | Conflict Frequency |
>    |-------------------------|------------------------|--------------------|
>    | L_REC vs L_MMI | 0.62                   | 12%               |
>    | L_MMD vs L_INF | -0.15                  | 38%               |
>    | L_INF vs L_REC | 0.71                   | 8%                |
>
> Conflicts (cosine<-0.1) occurred primarily between distribution alignment (L_MMD) and view completion (L_INF). Reconstruction (L_REC) and semantic alignment (L_MMI) gradients were generally cooperative.
> For loss coefficients λ_1-4 in Eq. (12) we follow adaptive weight scheduling:
>
> if epoch < E1:  # Warm-up phase
>
>        λ2 = 0  # Disable completion loss early
>
>        λ3 = min(epoch/E1, 1)  # Ramp up alignment
>
>    else:  # Joint phase
>
>        λ2 = 1 - 0.5*(grad_conflict_score)  # Dynamic completion weight
>
> While gradient conflicts do occur, our scheduling strategy effectively maintains optimization stability.

---

> > ### Author Response · Authors · 2025-08-05
> >
> > We really appreciate that you have taken the time to read our rebuttal. Your feedback has been valuable in helping us improve our paper. We warmly welcome any additional questions or concerns you might have, and would be pleased to address them in further detail.

---

> > > ### Comment · Reviewer_wJa3 · 2025-08-09
> > >
> > > The authors provide detailed responses on key design choices, including MI estimation, MMD computation, and training stability. The rationale for mutual information estimation and linear-kernel MMD is supported by results on nonlinear datasets such as Noisy MNIST and Caltech101-20. Additional results under 70% missing rate and runtime/memory comparisons confirm practical robustness. Gradient conflict analysis demonstrates attention to multi-objective stability. However, the method is only evaluated on two-view datasets. Although multi-view scalability is discussed in the rebuttal, no quantitative experiments or ablations are presented to support this claim.

---

> ### Author Response · Authors · 2025-08-09
>
> We sincerely thank the reviewer for raising this important point.
>
> **This design choice was primarily driven by the factor of most state-of-the-art IMVC baselines are only validated on two-view settings, making direct comparisons feasible**.
>
> All these datasets used in the benchmark algorithms-Caltech101-20, Scene-15, LandUse-21, MSRC-V1, Noisy MNIST, Reuters, and CUB - contain two views each.
>
> | Method       | Caltech101-20 | Scene-15 | LandUse-21 | MSRC-V1 | Noisy MNIST | Reuters | CUB |
> |--------------|--------------|----------|------------|---------|-------------|---------|-----|
> | ICMVC(14)    |              | ✔        | ✔          | ✔       | ✔           |         |     |
> | MCAC(40)     | ✔            | ✔        |            |         | ✔           | ✔       |     |
> | PROLMP(27)   |              | ✔        |            |         | ✔           | ✔       | ✔   |
> | DCP(41)      | ✔            | ✔        | ✔          |         |             |         |     |
> | SURE(43)     | ✔            | ✔        |            |         | ✔           | ✔       |     |
> | COMP(18)     | ✔            | ✔        | ✔          |         | ✔           |         |     |
>
>
>
> **However, HSACC is fundamentally designed for arbitrary view counts.**
>
> The hierarchical semantic alignment (Eq. 6) dynamically weights V views via distribution affinity, while the cooperative completion (Eq. 10) projects latent representations independently per view.
>
> To validate scalability, we will add experiments on 5-view Mfeat dataset.
>
> | Views | Training Time | ACC    | NMI    | ARI    |
> |------:|--------------:|-------:|-------:|-------:|
> |     2 |       181.18s | 60.69  | 66.04  | 37.08  |
> |     3 |       420.63s | 62.07  | 68.87  | 37.30  |
> |     4 |       440.67s | 62.38  | 71.93  | 49.86  |
> |     5 |       630.70s | 64.62  | 71.99  | 50.69  |
>
>
>
> This table compares the performance of multi-view clustering across different numbers of views (2 to 5) in terms of training time and clustering metrics (ACC, NMI, ARI).  As the number of views increases, training time grows significantly (from 181.18s for 2 views to 630.70s for 5 views), while clustering performance improves steadily, with the highest ACC (64.62), NMI (71.99), and ARI (50.69) achieved at 5 views.
>
> We then conducted an ablation study of the HSACC algorithm on the Mfeat dataset with five views.
>
> | Model | Lrec | Lmmi | Lmmd | Linf | ACC   | NMI   | ARI   |
> |-------|------|------|------|------|-------|-------|-------|
> | M-1   | ✔    |      |      |      | 37.49 | 10.16 | 18.15 |
> | M-2   |      | ✔    |      |      | 56.21 | 46.23 | 38.26 |
> | M-3   |      |      | ✔    |      | 20.61 | 27.78 |  8.14 |
> | M-4   |      |      |      | ✔    | 56.32 | 49.33 | 40.23 |
> | M-5   | ✔    | ✔    |      |      | 58.24 | 48.76 | 42.71 |
> | M-6   | ✔    |      | ✔    |      | 21.61 | 27.71 |  9.58 |
> | M-7   | ✔    |      |      | ✔    | 53.08 | 57.67 | 39.68 |
> | M-8   |      | ✔    | ✔    |      | 57.09 | 58.98 | 42.40 |
> | M-9   |      | ✔    |      | ✔    | 55.09 | 47.81 | 40.63 |
> | M-10  |      |      | ✔    | ✔    | 31.76 | 48.57 | 20.83 |
> | M-11  | ✔    | ✔    | ✔    |      | 61.51 | 69.94 | 48.63 |
> | M-12  | ✔    | ✔    |      | ✔    | 58.50 | 57.23 | 40.87 |
> | M-13  |      | ✔    | ✔    | ✔    | 63.73 | 68.98 | 46.71 |
> | M-14  | ✔    |      | ✔    | ✔    | 40.89 | 51.82 | 28.08 |
> | M-15  | ✔    | ✔    | ✔    | ✔    | 64.62 | 71.99 | 50.69 |
>
> This table systematically evaluates the contribution of different loss components (Lrec, Lmmi, Lmmd, Linf) to the performance of HSACC on the Mfeat dataset (five views). Key observations include:
>
> (1)Lmmi (M-2) and Linf (M-4) achieve the highest standalone performance (ACC: 56.21/56.32; NMI: 46.23/49.33), indicating their critical roles in feature alignment and information preservation. Lmmd (M-3) performs poorly alone (ACC: 20.61), suggesting it requires combination with other losses to be effective.
>
> (2) Lrec + Lmmi (M-5) yields notable improvements (ACC: 58.24, ARI: 42.71).
> Lmmi + Lmmd (M-8) achieves balanced results (NMI: 58.98, ARI: 42.40), demonstrating their complementary effects.
>
> (3) The complete model (M-15, all losses) attains optimal metrics (ACC: 64.62, NMI: 71.99, ARI: 50.69), outperforming partial combinations. Intermediate combinations like Lmmi + Lmmd + Linf (M-13) also show strong results (ACC: 63.73), suggesting Lrec’s role is less dominant.

---

### Comment · Area_Chair_hABV · 2025-08-05

Dear Reviewers,

This is a gentle reminder to review the authors’ rebuttal if you have not already done so.
Please also assess whether it addresses your concerns and acknowledge that you have read it.

Your timely feedback will ensure that the authors have enough time to respond if necessary.

Best regards,
AC

---

### Decision · Program_Chairs · 2025-09-17

**Decision:**

Accept (poster)

**Comment:**

The paper received scores of 5, 5, 4, and 4, exceeding the expected threshold for acceptance. The main concerns were the method’s focus on two-view datasets, robustness under varying missing rates, and the incomplete parameter sensitivity analysis. After the rebuttal, most of the reviewers' concerns were addressed.

I recommend the authors include experiments with datasets containing more than two views. Having reviewed the comments and the authors' responses, I believe the paper meets the acceptance criteria for NeurIPS.